# Unveiling the signaling network of FLT3-ITD AML improves drug sensitivity prediction

Sara Latini[1†], Veronica Venafra[1†], Giorgia Massacci[2], Valeria Bica[1], Simone Graziosi[1], Giusj Monia Pugliese[2], Marta Iannuccelli[2], Filippo Frioni[3], Gessica Minnella[4], John Donald Marra[3], Patrizia Chiusolo[3], Gerardo Pepe[2], Manuela Helmer Citterich[2], Dimitros Mougiakakos[5,6], Martin Böttcher[5,6], Thomas Fischer[5,7], Livia Perfetto[2,8*], Francesca Sacco[2,9*]

[1]Cellular and Molecular Biology, Department of Biology, University of Rome Tor Vergata, Rome, Italy; [2]Department of Biology, University of Rome Tor Vergata, Rome, Italy; [3]Sezione di Ematologia, Dipartimento di Scienze Radiologiche ed Ematologiche, Università Cattolica del Sacro Cuore, Rome, Italy; [4]Dipartimento di Diagnostica per Immagini, Radioterapia Oncologica ed Ematologia, Fondazione Policlinico A. Gemelli IRCCS, Rome, Italy; [5]Health Campus for Inflammation, Immunity and Infection (GCI3), Otto-von-Guericke University of Magdeburg, Magdeburg, Germany; [6]Department of Hematology and Oncology, Otto-von-Guericke University of Magdeburg, Magdeburg, Germany; [7]Institute of Molecular and Clinical Immunology, Otto-von-Guericke University of Magdeburg, Magdeburg, Germany; [8]Department of Biology, Fondazione Human Technopole, Milan, Italy; [9]Telethon Institute of Genetics and Medicine (TIGEM), Pozzuoli, Italy

**\*For correspondence:**
livia.perfetto@uniroma1.it (LP);
francesca.sacco@uniroma2.it (FS)

[†]These authors contributed equally to this work

**Abstract** Currently, the identification of patient-specific therapies in cancer is mainly informed by personalized genomic analysis. In the setting of acute myeloid leukemia (AML), patient-drug treatment matching fails in a subset of patients harboring atypical internal tandem duplications (ITDs) in the tyrosine kinase domain of the FLT3 gene. To address this unmet medical need, here we develop a systems-based strategy that integrates multiparametric analysis of crucial signaling pathways, and patient-specific genomic and transcriptomic data with a prior knowledge signaling network using a Boolean-based formalism. By this approach, we derive personalized predictive models describing the signaling landscape of AML FLT3-ITD positive cell lines and patients. These models enable us to derive mechanistic insight into drug resistance mechanisms and suggest novel opportunities for combinatorial treatments. Interestingly, our analysis reveals that the JNK kinase pathway plays a crucial role in the tyrosine kinase inhibitor response of FLT3-ITD cells through cell cycle regulation. Finally, our work shows that patient-specific logic models have the potential to inform precision medicine approaches.

## eLife assessment

This **important** study could potentially represent a step forward towards personalized medicine by combining cell-based data and a prior-knowledge network to derive Boolean-based predictive logic models to uncover altered protein/signaling networks within cancer cells. The level of evidence supporting the conclusions is **solid**, as the authors present analyses on independent, real-world data to validate their approach. These findings could be of interest to medical biologists working in the field of cancer, as the work should inform drug development and treatment choices in the field of oncology.

## Introduction

In the era of precision medicine, comprehensive profiling of malignant tumor samples is becoming increasingly time- and cost-effective in clinical ecosystems (*De Maria Marchiano et al., 2021*; *Tsimberidou et al., 2020*). While a growing number of genotype-tailored treatments have been approved for use in clinical practice (*Krzyszczyk et al., 2018*; *Scheetz et al., 2019*), the success of targeted therapies is limited by frequent development of drug resistance mechanisms that lead to therapy failure and portend a dismal patient prognosis (*Mansoori et al., 2017*; *Sabnis and Bivona, 2019*; *Vander Velde et al., 2020*; *Vasan et al., 2019*). Drug combinations are currently under investigation as a potential means of avoiding drug resistance and achieving more effective and durable treatment responses.

As the number of possible combinations increases exponentially with the number of drugs available, it is impractical to test for potential synergistic properties among all available drugs using empirical experiments alone. Computational approaches that can predict drug synergy, including Boolean logic models, are crucial in guiding experimental approaches for discovering rational drug combinations. In the Boolean model, a biological process or pathway of interest is modeled in the form of a signed and direct graphic with edges representing the regulatory relationship (activating or inhibitory) between the nodes (proteins). Logical operators (AND, OR, and NOT) are then employed to dynamically describe how the signal is integrated and propagated in the system over time to reach a terminal state. These states can be associated with cellular processes such as apoptosis and proliferation (*Calzone et al., 2022*). Once optimized, these models offer the ability to test for the effect of perturbation of the nodes on the resulting phenotype (e.g. in silico knock-out), allowing us to generate novel hypotheses and to predict the efficacy of novel drug combinations (*Hemedan et al., 2022*; *Le Novère, 2015*; *Montagud et al., 2022*; *Schwab et al., 2020*; *Wang et al., 2012*).

Among the different computational methods available, in the present study, we utilized CellNOptR (*Terfve et al., 2012*) to implement an integrated strategy that combines prior knowledge signaling networks (PKNs) with multiparametric analysis and Boolean logic modeling. We applied this approach to generate genotype-specific predictive models of AML patients with differing sensitivities to drug treatments. Specifically, we focused on a subset of AML patients with internal tandem duplication (ITDs) in the FLT3 receptor tyrosine kinase. FLT3-ITD, one of the most common driver mutations in AML, occurs in exons 15 and 16, which encode the juxtamembrane domain (JMD) and the first tyrosine kinase (TKD1) domain, and results in constitutive activation. We and others have demonstrated that the location (insertion site) of the ITD is a crucial prognostic factor: treatment with the recently FDA-approved multi-kinase inhibitor Midostaurin and standard frontline chemotherapy has a significant beneficial effect only in patients carrying the ITDs in the JMD domain, whereas no beneficial effect has been shown in patients carrying ITDs in the TKD region (*Rücker et al., 2022*; *Pugliese et al., 2023*; *Massacci et al., 2023*). Moreover, our group and others have demonstrated that the differences underlying tyrosine kinase inhibitor (TKI) sensitivity are related to a genotype-specific rewiring of the involved signaling networks.

In the present study, we applied a newly developed integrated approach to construct predictive logic models of cells expressing FLT3[ITD-TKD] and FLT3[ITD-JMD]. These models revealed that JNK plays a crucial role in the TKI response of FLT3-ITD cells through a cell cycle-dependent mechanism, in line with our previous findings (*Massacci et al., 2023*; *Pugliese et al., 2023*). Additionally, we integrated patient-specific genomic and transcriptomic data with cell line-derived logic models to obtain predictive personalized mathematical models with the aim of proposing novel patient-individualized anti-cancer treatments.

## Results

### The experimental strategy

In the treatment of cancer, molecular-targeted therapies often have limited effectiveness, as tumors can develop resistance over time. One potential solution to this problem is the use of combination therapy, for which data-driven approaches are valuable in identifying optimal drug combinations for individual patients. To identify novel genotype-specific combinatorial anti-cancer treatments in AML patients with FLT3-ITD, we employed a multidisciplinary strategy combining multiparametric analysis with literature-derived causal networks and Boolean logic modeling. Our experimental model

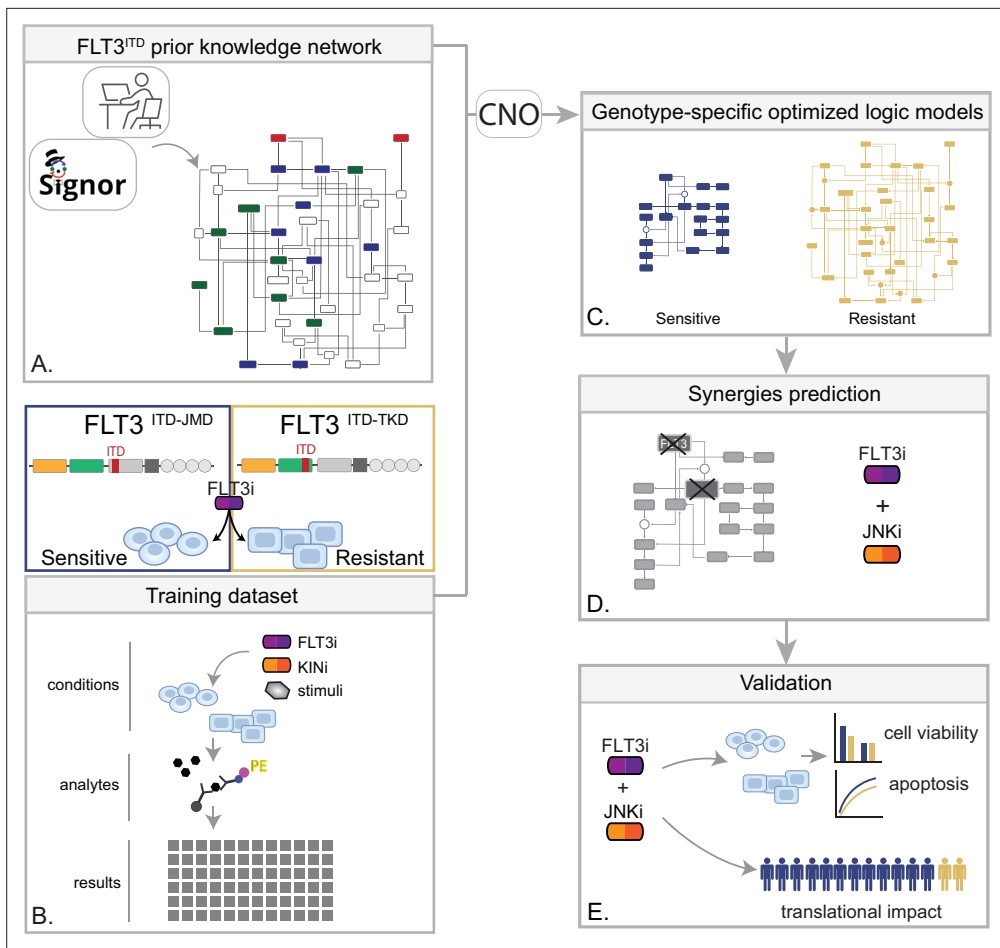

**Figure 1.** Summary of the experimental strategy. (**A**) Manual curation of FLT3-internal tandem duplication (ITD)-specific prior knowledge network (PKN). (**B**) Multiparametric analysis of signaling perturbations in tyrosine kinase inhibitor (TKI)-sensitive and TKI-resistant cells. (**C**) Model generation through the CellNOptR tool. (**D**) Prediction of combinatorial treatments restoring TKI sensitivity. (**E**) In vitro validation of novel combinatorial treatments.

consisted of hematopoietic Ba/F3 cells stably expressing the FLT3 gene with ITDs insertions in the JMD domain (aa 598) or in the TKD1 (aa 613) region (henceforth 'FLT3^ITD-JMD' and 'FLT3^ITD-TKD' cells, respectively). As previously demonstrated, cells expressing FLT3^ITD-TKD ('resistant model') have significantly decreased sensitivity to TKIs, including the recently registered FLT3-TKI Midostaurin, compared with FLT3^ITD-JMD cells ('sensitive model') (*Massacci et al., 2023*; *Pugliese et al., 2023*). Our approach (schematized in *Figure 1*) is summarized as follows:

1. Step 1: The first step in our strategy aimed at providing a detailed description of FLT3-ITD-triggered resistance mechanisms. To this end, we carried out a curation effort and mined our in-house resource, SIGNOR (*Lo Surdo et al., 2023*), to build a PKN recapitulating known signaling pathways downstream of the FLT3 receptor. The PKN integrates information obtained in different cellular systems under distinct experimental conditions (*Figure 1A*).
2. Step 2: Using the PKN, we selected 14 crucial proteins, which we refer to as 'sentinel proteins', whose protein activity was emblematic of the cell state downstream of FLT3. Thus, by performing a multiparametric analysis, we measured the activity status of the sentinel proteins under 16 different perturbation conditions in TKIs sensitive and resistant cells to generate the training dataset (*Figure 1B*).
3. Step 3: We employed the CellNOptR tool to optimize the PKN using the training data. Two genotype-specific predictive models were generated that best reproduced the training dataset (*Figure 1C*).

4. Step 4: Using the optimized model, we performed an in silico knock-out screen involving the suppression of multiple crucial nodes. Novel combinatorial treatments were predicted according to the induction of apoptosis in TKI-sensitive and TKI-resistant cells (*Figure 1D*).

5. Step 5: The predictive performance of the two models was validated in vitro, and in silico in two independent publicly available datasets. The clinical impact of our models was assessed in a cohort of 14 FLT3-ITD positive AML patients (*Figure 1E*).

## Generation of FLT3-ITD PKN

The first step in the application of our pipeline consisted of the creation of the PKN, a static and genotype-agnostic map recapitulating the signaling pathways deregulated over AML tumor development and progression (*Figure 2—figure supplement 1*). To create the PKN, we embarked on a curation effort aimed at describing the molecular mechanisms or causal relationships connecting three crucial receptors responsible for sustaining the proliferative and survival pathways in AML (FLT3, TNFR, and IGF1R), to downstream events (i.e. apoptosis and proliferation). Gathered data were captured using our in-house developed resource, SIGNOR, and made freely accessible to the community for reuse and interoperability, in compliance with the FAIR principles (*Wilkinson et al., 2016*). Briefly, SIGNOR is a public repository that captures more than 35K causal interactions (up/down-regulations) among biological entities and represents them in the form of a direct and signed network (*Lo Surdo et al., 2023*). This representation format makes it particularly suitable for the implementation of Boolean logic modeling approaches. The so-obtained pre-PKN included 76 nodes and 193 edges, the nodes representing proteins, small molecules, stimuli, and phenotypes, and the edges depicting the directed interactions between the nodes (*Supplementary file 1*).

As little is known about the specific signaling pathways downstream of the non-canonical FLT3$^{ITD-TKD}$, we enriched the pre-PKN, deriving new edges from cell-specific experimental data of both FLT3$^{ITD-TKD}$ and FLT3$^{ITD-JMD}$ expressing cell lines (see Materials and methods, *Figure 2A*). This refined PKN recaps the FLT-ITD downstream signaling and served as the basis for the model optimization.

## Multiparametric analysis of TKI-resistant and -sensitive FLT3-ITD cells

To clarify the cooperative and antagonistic interactions among FLT3 inhibition and complementary therapeutic strategies, we generated a cue-sentinel-response multiparametric dataset (*Supplementary file 2*, *Supplementary file 3*, and *Supplementary file 4*). Generally, MAPKs, PI3K-AKT-mTOR, and STATs are the main pathways downstream of FLT3, IGF1R, and TNFR; we selected six key kinases in the FLT3-ITD PKN, and we perturbed their activity with small molecule inhibitors in presence or in absence of the FLT3i Midostaurin. We added the cytokines as stimuli to fully activate the RTKs included in our network (*Figure 2B*). Specifically, we treated sensitive and resistant cells with either PI3Ki, mTORi, MEKi, or GSKi±Midostaurin for 90 min and we added IGF1 for the last 10 min. Parallely, we treated the cells with p38i or JNKi±Midostaurin for 90 min and added TNFα for the last 10 min (*Figure 2C*). Overall, we subjected our cell lines to 16 experimental conditions (listed in Materials and methods, *Supplementary file 2*) and in each of them we measured the signaling perturbations. As sentinels of the signaling activity response, we measured in triplicate the activity states of 14 crucial proteins (*Figure 2B and C*) based on their phosphorylation status (mTOR, CREB1, IGF1R, PTEN, GSK3a, GSK3b, STAT3, STAT5, TSC2, p70S6K, RPS6, JNK, p38, ERK1/2).

Briefly, the biological replicates displayed Pearson correlation coefficients ranging between 0.75 and 1 (*Figure 2—figure supplement 2A and B*). Overall, the observed modulation of the readouts was consistent with the experimental evidence reported in the literature (*Figure 2—figure supplement 2C*, black squares in the heatmap). For each sentinel protein, we employed combinations of inhibitors and stimuli to probe the full spectrum of protein activity, ranging from the minimum (inhibitor treatment) to the maximum (stimulus exposure). The data were normalized in the 0–1 range using a Hill function. In this way, the fully active sentinel value was = 1, and the inhibited value = 0 (*Figure 2—figure supplement 3*).

Principal component analysis (PCA) (*Figure 2D*) and unsupervised hierarchical clustering (*Figure 2—figure supplement 2D*) showed that the activity level of sentinel proteins stratified cells according to both FLT3 activation status (component 1: presence vs absence of FLT3i) and cytokine stimulation (component 2: IGF1 vs TNFα). Of note, among all the KINi-treated conditions, only the JNKi treatment groups with the FLT3i treated samples in both cell lines. On the contrary, the activation profile of

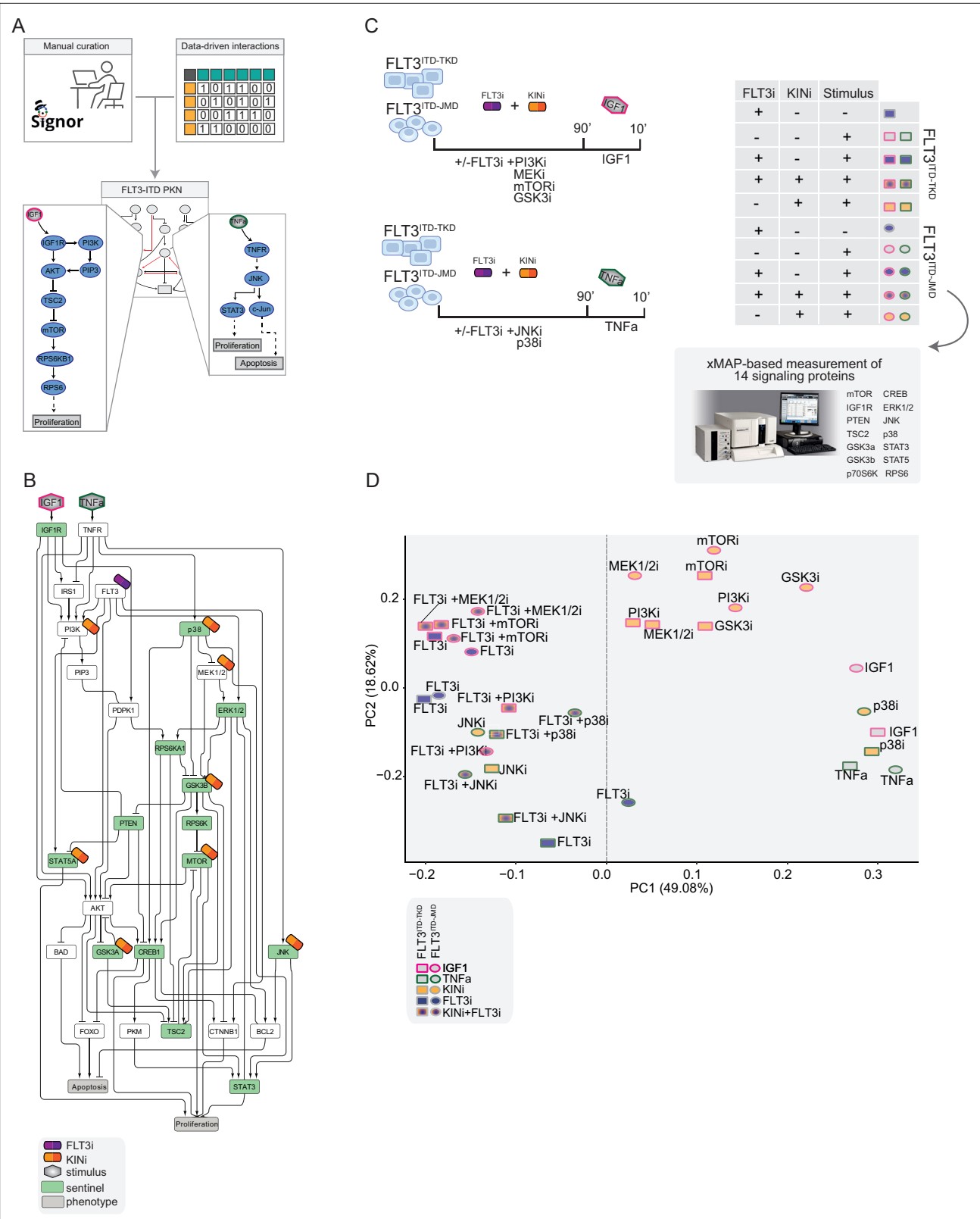

**Figure 2.** Generation of the training dataset. (**A**) Schematic representation of the FLT3-internal tandem duplication (ITD) prior knowledge network (PKN) manual curation, integration of data-driven edges, and manual integration of RTKs pathways involved in acute myeloid leukemia (AML). (**B**) Schematic representation of the experimental design: FLT3$^{ITD-JMD}$ and FLT3$^{ITD-TKD}$ cells were cultured in starvation medium (w/o FBS) overnight and treated with PI3Ki, MEKi, mTORi, and GSK3i, JNKi and p38i, in presence or absence of the FLT3i Midostaurin for 90 min. Then, the cells were stimulated either with

*Figure 2 continued on next page*

*Figure 2 continued*

IGF1 or TNFα for 10 min. Control cells were starved and treated with Midostaurin for 90 min. After treatment, samples were collected, and cell lysates were analyzed with an xMAP-based assay through the MagPix instrument. Per each experimental condition, we measured the phosphorylation levels of 14 sentinels. (**C**) Network representation of a compressed PKN that shows the essential pathways monitored through the perturbation experiment. The perturbed nodes are tagged with a drug icon, and the measured nodes are colored green. (**D**) Principal component analysis (PCA) of FLT3$^{ITD-JMD}$ and FLT3$^{ITD-TKD}$ cells in the multiparametric analysis. Each point represents a different experimental condition.

The online version of this article includes the following figure supplement(s) for figure 2:

**Figure supplement 1.** FLT3-internal tandem duplication (ITD) manually curated prior knowledge network (PKN).

**Figure supplement 2.** Global overview of multiparametric data.

**Figure supplement 3.** Normalization of analytes activity through Hill curves.

these 14 sentinel proteins was not able to distinguish cells according to the distinct FLT3-ITD insertion sites (circles = FLT3$^{ITD-JMD}$ and squares = FLT3$^{ITD-TKD}$) (*Figure 2D*). Interestingly, the unsupervised hierarchical clustering of the 14 analytes revealed different groups according to the pathway proximity of the nodes (e.g. JNK-p38; p70S6K-RPS6; STAT3-STAT5) or according to their regulatory role (e.g. GSK3a/b, PTEN and TSC2, acting as negative regulators, cluster together) (*Figure 2—figure supplement 2E*). Together, these observations suggest that the characterization of the genotype-dependent rewiring of signaling pathways cannot be obtained by simply looking at single proteins in our multiparametric dataset, but rather requires a modeling approach.

## Generation of FLT-ITD optimized logic models

CellNOptR was used to derive biologically relevant information from our dataset and generate FLT3-ITD-specific predictive models. Boolean logic models were optimized by maximizing the concordance between the PKN and our cue-sentinel-response multiparametric training dataset (*Figure 3A*).

In the first step, CellNOptR preprocesses the PKN (*Figure 2—figure supplement 1*) and translates it into logical functions (scaffold model). As previously described (*Sacco et al., 2012*), the preprocessing consists of three phases: (i) *compression*, in which unmeasured and untargeted proteins, as well as linear cascades of undesignated nodes, are removed; (ii) *expansion*, in which the remaining nodes are connected to the upstream regulators with every possible combination of OR/AND Boolean operators; and (iii) *imputation*, in which the software integrates the scaffold model with regulations function inferred without bias from the training dataset.

Using this strategy, we obtained two FLT3-ITD-specific PKNs, accounting for 206 and 208 nodes and 756 and 782 edges, for FLT3$^{ITD-JMD}$ and FLT3$^{ITD-TKD}$, respectively. The variation in the node count between these two PKNs results from the inclusion of a different number of AND Boolean operators during the *expansion* step, while the difference in edge numbers is primarily due to different variations in the data, leading to distinct edge connections in the *imputation* step.

In the second step, causal paths and Boolean operators from the scaffold models were filtered to best fit the experimental context (see Materials and methods). Briefly, for each cell line, we trained the software with our normalized cue-sentinel-response multiparametric dataset to generate a family of 1000 optimized Boolean models, and we retained the top 100 performing models (*Figure 3—figure supplement 1A*).

To qualitatively assess the robustness and reliability of the selected models, we compared the average activity modulation of the individual sentinel proteins with experimentally observed readouts (*Figure 3A*, panels 1–2).

Since the performance of the model strongly depends on the topology of the PKN, we performed several rounds of PKN check and adjustment, and, in each round, the entire process was iterated until the simulation provided the best fit of the available data (*Figure 3—figure supplement 1B and C*). As shown in *Figure 3A*, panel 3, the fit between simulated and experimental data was generally higher in the FLT3$^{ITD-JMD}$ model, which has been more extensively characterized by the scientific community than the FLT3$^{ITD-TKD}$ system. For each cell line, we selected the model with the lowest error (see Materials and methods) between experimental and simulated data in the two cell lines (best model) (*Figure 3B and C*). Interestingly, the two Boolean models display a different structure, and most of the interactions are cell-specific (blue edges), with only a few edges shared among the two networks (e.g. TNFR-FLT3,

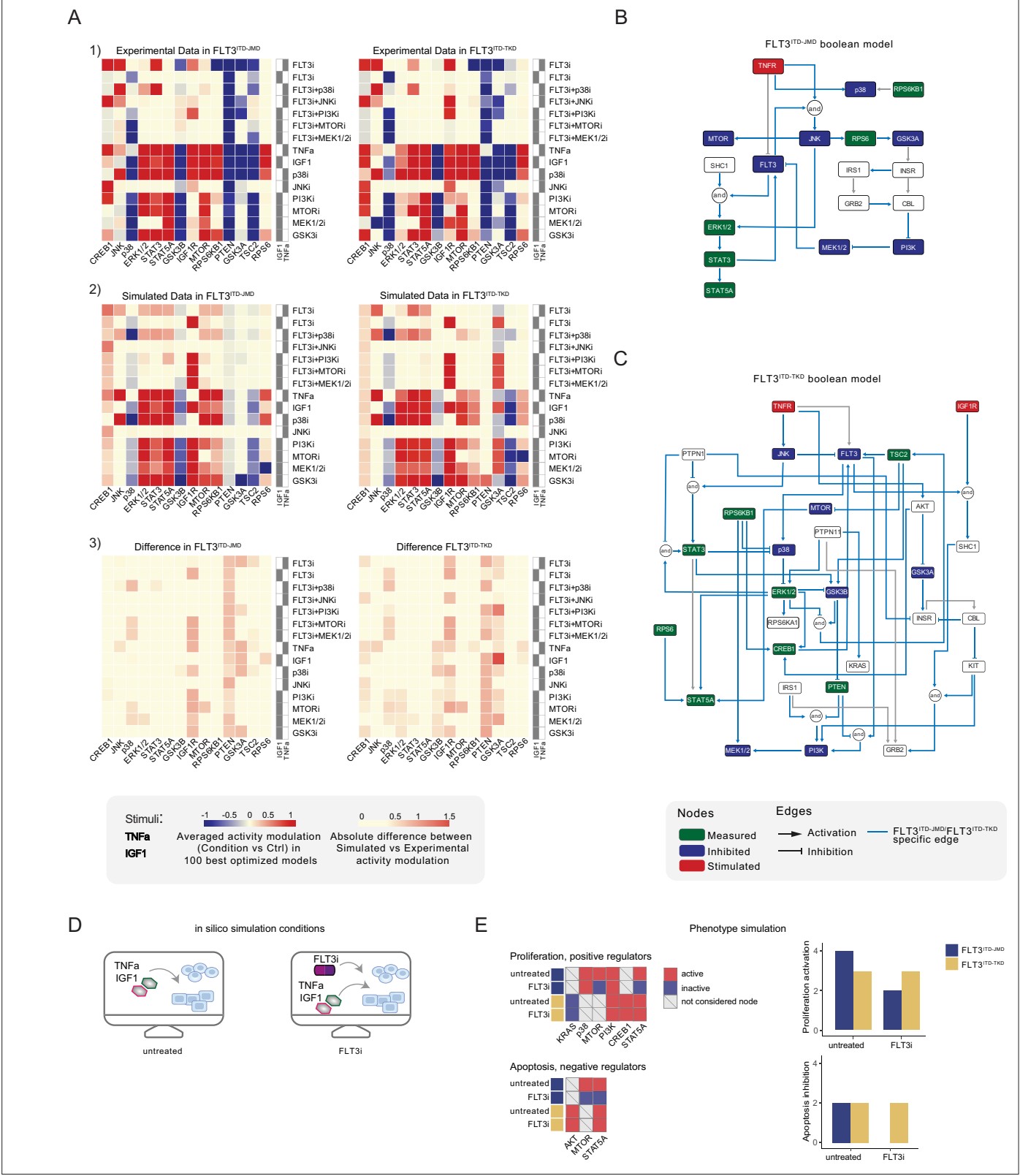

**Figure 3.** Optimized Boolean models recapitulate the different sensitivity of FLT3^ITD-JMD and FLT3^ITD-TKD cells to tyrosine kinase inhibitor (TKI). (**A**) Color-coded representations of the experimental activity modulation (T90–T0) of sentinel proteins used to train the two Boolean models (*upper panel*) and the average prediction of protein activities in the family of 100 best models (*central panel*). The protein activity modulation ranges from –1 to 1 and is represented with a gradient from blue (inhibited) to red (activated). The absolute value of the difference between experimental and simulated protein

Figure 3 continued

activity modulation (*lower panel*) is reported as a gradient from light yellow (error <0.5) to red (1.85). (**B–C**) FLT3[ITD-JMD] (**B**) and FLT3[ITD-TKD] (**C**) high-confidence Boolean models. Perturbed proteins in the experimental setup are marked in red or green if inhibited or stimulated, respectively. Sentinel proteins are reported in blue. The edges' weight represents their frequency in the family of 100 models and only the high-confidence ones (frequency >0.4) are reported. Orange edges are cell-specific links. (**D**) Cartoon of the in silico conditions simulated to analyze the different TKI sensitivity of the FLT3[ITD-JMD] and FLT3[ITD-TKD] Boolean models. Untreated condition: TNFα+IGF1; FLT3i: TNFα+IGF1+FLT3 inhibition. (**E**) Heatmaps (left) report the activation level of positive and negative phenotype regulators present in the two Boolean models. Bar plots (right) showing the proliferation activation and apoptosis inhibition levels in untreated and FLT3i conditions in the steady states of FLT3[ITD-JMD] (blue) and FLT3[ITD-TKD] (yellow) Boolean models.

The online version of this article includes the following figure supplement(s) for figure 3:

**Figure supplement 1.** Overview of the optimized Boolean models.

**Figure supplement 2.** Validation of FLT3-internal tandem duplication (ITD)-specific models on real-world independent datasets.

STAT3-STAT5A, p70S6K-p38, etc.). The architectural differences between the models demonstrate a profound rewiring of the signal downstream of FLT3 as a result of the different locations of the ITD.

## Evaluating the predictive power of FLT3-ITD logic models

Thus, we first took advantage of the publicly available quantitative phosphoproteomics dataset to independently validate our models. To this aim, we computed the steady state of the two models in 'untreated' and 'FLT3i' conditions (*Figure 3—figure supplement 1D*). Briefly, the *untreated* condition represents the tumor state, here the pro-survival receptors (FLT3, IGF1R, and TNFR) are set constitutively active and assigned a Boolean value of 1. In the *FLT3i* condition (Midostaurin administration), the FLT3 receptor is inhibited and associated with a Boolean value of 0, whereas IGF1R and TNFR remain constitutively active to reflect the environmental background that sustains tumor growth and proliferation (*Figure 3D*). Given these two initial conditions (*untreated* vs *FLT3i*), we carried out a synchronous simulation (*Schwab et al., 2020*) to compute the evolution of the two models. Next, we compared the steady state of our model upon FLT3 inhibition with the phosphoproteomic data describing the modulation of 16,319 phosphosites in FLT3-ITD Ba/F3 cells (FLT3[ITD-TKD] and FLT3[ITD-JMD]) upon quizartinib (AC220) treatment (*Massacci et al., 2023*). The activation status of the nodes in the two generated models is highly comparable with the level of regulatory phosphorylations reported in the reference dataset, supporting our models (*Figure 3—figure supplement 2A*). Next, we aimed to assess whether the newly generated FLT3[ITD-JMD] and FLT3[ITD-TKD] Boolean models could recapitulate in silico the modulation of apoptosis and proliferation upon inhibition of FLT3 and other druggable nodes of our models. First, to functionally interpret the results of the simulations, for each network, we extracted key regulators of 'apoptosis' and 'proliferation' hallmarks from SIGNOR. To this aim, we applied our recently developed ProxPath algorithm, a graph-based method able to retrieve significant paths linking the nodes of our two optimized models to proliferation and apoptosis phenotypes (*Iannuccelli et al., 2023*; see Materials and methods; *Supplementary file 1*; *Figure 3E*, left panels; *Figure 3—figure supplement 1D*). Then, integrating the signal of their key regulators (see Materials and methods), we were able to derive the 'proliferation' and 'apoptosis inhibition' levels upon each initial condition.

Importantly, our strategy demonstrated that FLT3[ITD-JMD] and FLT3[ITD-TKD] Boolean models were able to recapitulate the different TKI sensitivity of FLT3-ITD cells (*Figure 3E*, right panels) (*Massacci et al., 2023*; *Pugliese et al., 2023*).

Moreover, by taking advantage of the Beat AML program, which provides ex vivo drug sensitivity screening data of 134 FLT3[ITD-JMD] AML patients, we validated the prediction power of our models by comparing our in silico results with the in vitro IC50 values measured upon RTKs inhibition (*Figure 3—figure supplement 2B and C*). We observed some discrepancies between model's prediction and patients' data for PI3K inhibition (probably due to missing connections in our cell-specific model) while FLT3, mTOR, JNK, and p38 treatment outcomes in patients were successfully predicted by our models.

## Identification of novel combinatorial treatments reverting TKI resistance

As per their intrinsic nature, the two optimized Boolean logic models have predictive power and can be used to simulate in silico novel combinatorial treatments reverting drug resistance of FLT3<sup>ITD-TKD</sup> cells (*Figure 4A*).

Thus, we performed a targeted in silico approach in FLT3<sup>ITD-TKD</sup> and FLT3<sup>ITD-JMD</sup> cells, by simulating the levels of apoptosis and proliferation, upon combinatorial knock-out of FLT3 and one of the following key druggable kinases: ERK1/2, MEK1/2, GSK3A/B, IGF1R, JNK, KRAS, MEK1/2, mTOR, PDPK1, PI3K, p38. Interestingly, in the FLT3<sup>ITD-TKD</sup> model, the combined inhibition of JNK and FLT3, exclusively, in silico restores the TKI sensitivity, as revealed by the evaluation of the apoptosis and proliferation levels (*Figure 4B and C*).

We thus tested in vitro whether the pharmacological suppression of JNK using a highly selective inhibitor could increase the sensitivity of FLT3<sup>ITD-TKD</sup> cells to TKI treatment. Our data indicate that JNK plays a crucial role in cell survival of FLT3-ITD cells, since its pharmacological inhibition (SP600125) alone or in combination with Midostaurin (PKC412) significantly increased the percentage of apoptotic FLT3<sup>ITD-TKD</sup> cells (*Figure 4D*). Remarkably, the apoptosis of FLT3<sup>ITD-TKD</sup> patients-derived blasts is increased upon pharmacological inhibition of JNK (*Figure 4F*). Consistently, in these experimental conditions, we observed a significant reduction of proliferating FLT3<sup>ITD-TKD</sup> cells versus cells treated with Midostaurin alone (*Figure 4E*). Additionally, in agreement with the models' predictions, we demonstrated that pharmacological suppression of ERK1/2 or p38 kinases have no impact on the TKI sensitivity of FLT3<sup>ITD-TKD</sup> cells (*Figure 4—figure supplement 1A and B*).

We next sought to characterize the functional role of JNK in this response. Recently, we revealed that the cell cycle controls the FLT3<sup>ITD-TKD</sup> TKI resistance via the WEE1-CDK1 axis (*Massacci et al., 2023*). Interestingly, JNK has already been shown to play a role in cell cycle regulation through the inactivation of CDC25C, a phosphatase and positive regulator of CDK1 (*Gutierrez et al., 2010*). Thus, we investigated whether pharmacological inhibition of JNK may differently impact CDK1 activity in FLT3<sup>ITD-JMD</sup> and FLT3<sup>ITD-TKD</sup> cells. In line with our previous findings (*Massacci et al., 2023*; *Pugliese et al., 2023*), in FLT3<sup>ITD-JMD</sup> cells, Midostaurin treatment increases the dephosphorylated, cytosolic, and monomeric pool of CDK1 and inactivates CDK2 (*Figure 4G*), leading to cell accumulation in the G1 phase (*Figure 4H*). Combined treatment of SP600125 and Midostaurin increases CDK1 and CDK2 phosphorylation and CyclinB1 levels, increasing the percentage of G2-M and S-phases cells, compared with Midostaurin treatment alone (*Figure 4G and H*).

As expected, in cells expressing FLT3<sup>ITD-TKD</sup>, Midostaurin treatment triggers the formation of the inactive stockpiled pre-M-phase promoting factor (*Massacci et al., 2023*), constituted by the CDK1-CyclinB1 complex (*Figure 4G and I*). This complex is associated with a significant accumulation of proliferating FLT3<sup>ITD-TKD</sup> cells in the G2-M phase as compared to Midostaurin-treated FLT3<sup>ITD-JMD</sup> cells. In line with these observations, CDK2 phosphorylation on activating Thr160 was significantly increased (*Figure 4G*, *Figure 4—figure supplement 1E*). On the other hand, combined treatment of SP600125 and Midostaurin induces dephosphorylation of CDK1 on the inhibitory Tyr15 and a mild accumulation of FLT3<sup>ITD-TKD</sup> cells in G2-M phase (*Figure 4G and I*; *Figure 4—figure supplement 1C and D*; *Figure 4—figure supplement 1G*). These observations support the hypothesis that the combination of JNK inhibition with Midostaurin treatment impacts the cell cycle progression in TKI-resistant FLT3<sup>ITD-TKD</sup> cells, impairing their survival and reactivating the TKI-induced apoptosis.

## Generation of AML patient-specific logic models

Our genotype-specific Boolean models were built on in vitro signaling data and enabled us to formulate reliable mechanistic hypotheses underlying TKI resistance in our AML cellular models. As outlined in *Figure 5A*, to exploit their predictive power in a more clinical setting, we implemented a computational strategy that combines the models' topological structure with patient-derived gene expression data.

As a pilot analysis, we analyzed the mutational and expression profiles of 262 genes (*Supplementary file 7*), relevant to hematological malignancies in a cohort of 14 FLT3-ITD positive de novo AML patients (*Figure 5A*, panel a). Briefly, the classification of these 10 patients according to their ITD localization (see Materials and methods) was as follows: eight patients with FLT3<sup>ITD-JMD</sup>, four with FLT3<sup>ITD-JMD+TKD</sup>, and two with FLT3<sup>ITD-TKD</sup> (*Figure 5A*, panel b). The specific insertion sites of the ITD in

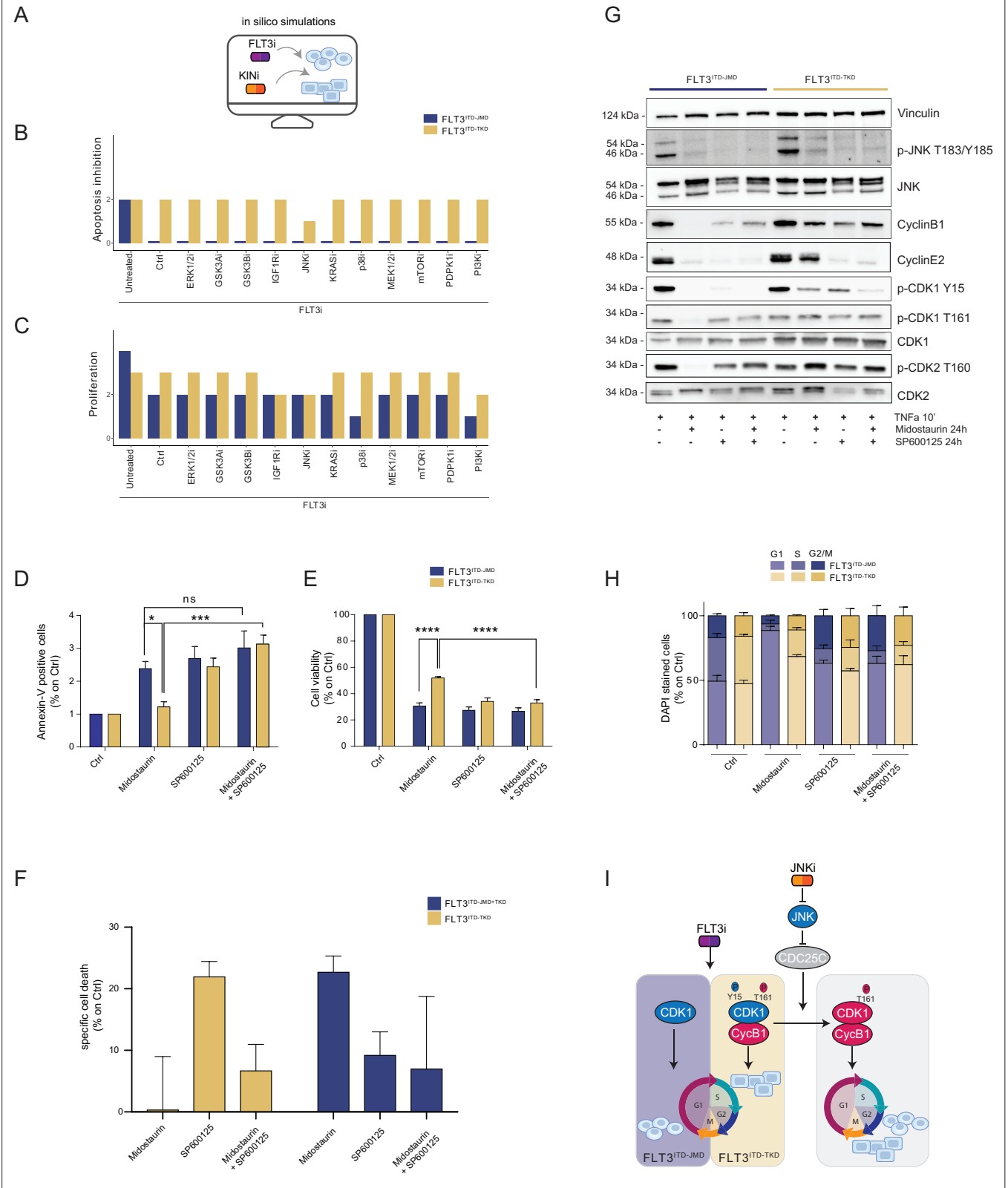

**Figure 4.** In silico simulation of the FLT3^ITD-TKD logic model allows the prediction of novel combinatorial treatment reverting tyrosine kinase inhibitor (TKI) resistance. (**A**) Cartoon of the in silico simulation conditions. (**B–C**) Bar plot showing the in silico simulation of proliferation activation (**B**) and apoptosis inhibition (**C**) levels in untreated and FLT3i conditions in combination with knock-out of each of 10 crucial kinases in FLT3^ITD-JMD (blue) and ^-TKD (yellow) cells. (**D–E**) In FLT3^ITD-JMD (blue) and ^-TKD (yellow) cells treated with 100 nM Midostaurin and/or 10 μM SP600125 (JNK inhibitor) for 24 hr, the percentage

*Figure 4 continued on next page*

*Figure 4 continued*

of Annexin V positive cells (**D**) and the absorbance values at 595 nm (**E**), normalized on control condition, are reported in bar plots. (**F**) Primary samples from acute myeloid leukemia (AML) patients with the FLT3$^{ITD-TKD}$ mutation (n=2, yellow bars) or the FLT3$^{ITD-JMD/TKD}$ mutation (n=3, blue bars) were exposed to Midostaurin (100 nM, PKC412), and JNK inhibitor (10 μM, SP600125) for 48 hr, or combinations thereof. The specific cell death of gated AML blasts was calculated to account for treatment-unrelated spontaneous cell death. The bars on the graph represent the mean values with standard errors. (**G**) In FLT3$^{ITD-JMD}$ (blue) and FLT3$^{ITD-TKD}$ (yellow) cells treated with 100 nM Midostaurin and/or 10 μM SP600125, followed by 10' of TNFα 10 ng/ml, the protein levels of phospho-JNK (T183/Y185), JNK, phospho-CDK1 (Y15), phospho-CDK1 (T161), CDK1, phospho-CDK2 (T160), CDK2, CyclinB1, CycinE2, and Vinculin were evaluated by western blot analysis. (**H**) Cytofluorimetric cell cycle analysis of DAPI-stained FLT3$^{ITD-JMD}$ (blue) and FLT3$^{ITD-TKD}$ (yellow) cells treated with 100 nM Midostaurin and/or 10 μM SP600125 (JNK inhibitor) for 24 hr. (**I**) Cartoon of the molecular mechanism proposed for FLT3$^{ITD-JMD}$ and FLT3$^{ITD-TKD}$ cells.

The online version of this article includes the following source data and figure supplement(s) for figure 4:

**Source data 1.** Original files for the western blot analysis in *Figure 4G* of phospho-JNK(T183/Y185), JNK, phospho-CDK1 (Y15), phospho-CDK1 (T161), CDK1, phospho-CDK2(T160), CDK2, CyclinB1, CycinE2, and Vinculin.

**Source data 2.** PDF containing *Figure 4G* and original scans for the western blot analysis of phospho-JNK(T183/Y185), JNK, phospho-CDK1 (Y15), phospho-CDK1 (T161), CDK1, phospho-CDK2(T160), CDK2, CyclinB1, CycinE2, and Vinculin, with highlighted band.

**Figure supplement 1.** Quantification of western blot analysis of cell cycle proteins.

---

the patient cohort are shown in *Supplementary file 8*. Follow-up clinical data were available for 10 out of 14 patients (*Figure 5B*, *Supplementary file 9*).

Mutation profiling analysis of the patient cohort revealed a heterogeneity in the genetic background among patients and a high number of co-occurring genetic alterations (*Figure 5—figure supplement 1A and B*; *Supplementary file 9*). By computing the genes' z-scores with respect to each patient's gene expression distribution, we detected patient-specific up- or down-regulated transcripts (*Figure 5A*, panel a; *Supplementary file 9*).

Significantly, patients' unsupervised hierarchical clustering according to the mutational profile or according to the z-score distribution of the gene expression and PCA of the gene expression data was unable to stratify patients based on their FLT3-ITD subtypes (*Figure 5C and D*; *Figure 5—figure supplement 1C*).

At this point, we tested whether we could exploit the cell-derived Boolean models to generate personalized predictive models able to reproduce the clinical outcome of patients and then identify novel personalized combinatorial treatments.

To these aims, each patient's mutational profile (*Figure 5—figure supplement 1A and D*, *Supplementary file 9*) was first used to match the suitable cell-derived FLT3-ITD model and then exploited to set the initial condition and obtain 14 personalized Boolean models (*Figure 5A*, panel c). Next, for each patient, we performed a simulation of the following conditions in silico: (i) *untreated* state; (ii) *FLT3i* condition (see Materials and methods); and (iii) combination of FLT3i and inhibition with previously tested kinases. Importantly, our approach enabled us to obtain patient-specific predictive Boolean models able to describe the drug-induced signaling rewiring (*Figure 5F* and *Figure 5—figure supplement 2*) and to quantify 'apoptosis inhibition' and 'proliferation' levels (*Figure 5A*, panels d and e, *Figure 5E* and *Figure 5—figure supplement 1E*). The anti-proliferative and pro-apoptotic response to FLT3 inhibition (*Figure 5E*) of JMD1, JMD2, JMD3, JMD7, and JMD6 models was confirmed by follow-up clinical data that displayed a favorable outcome upon treatment (*Figure 5B*). In fact, in JMD1 patient, the sole FLT3 inhibition impairs the STAT3-STAT5 and JNK-MTOR axes and leads to an anti-tumoral phenotype (*Figure 5F*). Conversely, simulations of JMD5, JMD_TKD2, JMD_TKD3, and TKD1 models showed an opposite outcome with respect to real-life clinical observations (*Figure 5B*). For example, in our in silico model, the clinically responder TKD1 patient (*Figure 5B*) was resistant to all the tested combinatorial treatments, with a weak effect of PI3Ki on the pro-proliferative axis (*Figure 5E* and *Figure 5—figure supplement 1E*). One possible explanation is that the more complex mutational landscape of the TKD1 patient cannot be recapitulated by our scaffold model (*Figure 5—figure supplement 1D*). Interestingly and in line with our previous cell-line-based findings, JNK inhibition appeared to be a promising approach to alleviate the resistant phenotype of the clinically-not-responder TKD2 (*Figure 5B*), as revealed by the diminished levels of 'apoptosis inhibition' and 'proliferation' (*Figure 5E and F*). Our model suggests that the effect of combinatorial FLT3i and JNKi treatment increases AML cell death through the STAT3/STA5A axis (*Figure 5F*). Overall, this analysis

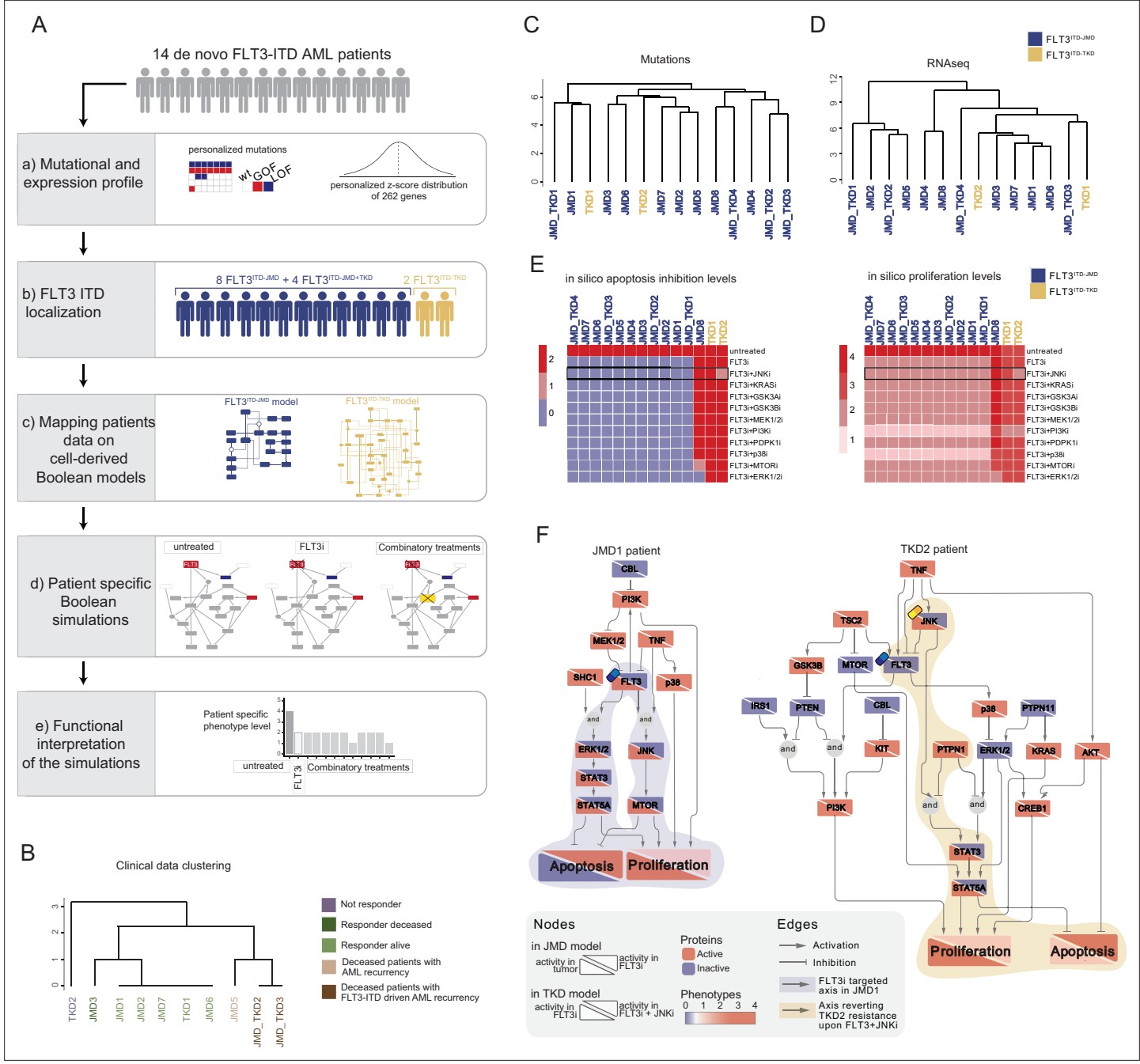

**Figure 5.** FLT3-internal tandem duplication (ITD) patient-specific Boolean models. (**A**) Schematic representation of our computational approach to obtain personalized logic models. (**B**) Hierarchical clustering of patients according to their clinical characteristics (response to chemotherapy, vital status, and acute myeloid leukemia [AML] recurrence). Resistant, alive or deceased responders, and deceased with general or FLT3-ITD AML recurrence patients are reported in purple, light or dark green, and light or dark brown, respectively. (**C–D**) Hierarchical clustering of patients according to their mutational profile (**C**) and their expression profile of 262 genes (**D**). (**E**) Heatmap representing patient-specific in silico apoptosis inhibition (*left panel*) and proliferation levels (*right panel*) upon each simulation condition. (**F**) Patient-specific (JMD1 and TKD2) Boolean models. In the JMD1 model (*left panel*), nodes' activity has been simulated in control (bottom-left part) and *FLT3 inhibition* conditions (upper-right part). In the TKD2 model (*right panel*), nodes' activity has been simulated in *FLT3 inhibition* (bottom-left part) and *FLT3 and JNK co-inhibition* conditions (upper-right part).

The online version of this article includes the following figure supplement(s) for figure 5:

**Figure supplement 1.** Overview of patient-specific Boolean models.

**Figure supplement 2.** Personalized Boolean models.

showcases the two trained Boolean logic models have predictive power and can contribute to identifying potential therapeutic strategies improving clinical outcome of FLT3[ITD-TKD] patients.

## Discussion

Cancer is primarily a signaling disease in which gene mutations and epigenetic alterations drastically impact crucial tumor pathways, leading to aberrant survival and cell proliferation. Indeed, nearly all molecularly targeted therapeutic drugs are directed against signaling molecules (*Min and Lee, 2022*). However, the success of targeted therapies is often limited, and drug resistance mechanisms arise, leading to therapy failure and dismal patient prognosis. To address this issue, a comprehensive, patient-specific characterization of signaling network rewiring can offer an unprecedented opportunity to identify novel promising, personalized combinatorial treatments.

Logic-based models have already been proven to successfully meet this challenge, thanks to their ability to condense the signaling features of a system and to infer the response triggered by genetic and chemical perturbations to the system in silico (*Lee et al., 2012*; *Montagud et al., 2022*).

In the present study, we optimized a methodology to investigate drug sensitivity using genotype-specific Boolean models. Our approach involved building a model representing the patient-specific cell state or disease status and inferring novel combinatorial anti-cancer treatments that may overcome drug resistance.

Here, we specifically applied this methodology to acute myeloid leukemia (AML) patients carrying the ITD in the FLT3 receptor tyrosine kinase. Our group recently showed, by integrating unbiased mass spectrometry-based phosphoproteomics with literature-derived signaling networks, that the location of the ITD insertion affects the sensitivity to TKIs therapy through a WEE1-CDK1-dependent mechanism. Our work enabled us to obtain a nearly complete, though static, picture of how FLT3-ITD mutations rewire signaling networks.

The main goal of the present study was the generation of clinically relevant, predictive models of the FLT3-ITD-dependent cell state. We aim to use these models to predict in silico the TKI sensitivity upon multiple simultaneous perturbations (i); to generate personalized models by combining patient-specific genomic and transcriptomic datasets (ii); and to propose novel, effective, patient-specific anti-cancer treatments (iii).

By taking advantage of the previously developed Cell Network Optimizer software (*Terfve et al., 2012*), we employed a multi-step strategy that trains a PKN with a large-scale multiparametric dataset by using a Boolean logic modeling formalism.

First, we generated a literature-derived FLT3-ITD-centered signaling network encompassing relevant pathways in AML, including the regulation of key phenotypes, such as apoptosis and proliferation. Manually curated data were made publicly available and can be freely explored by using tools offered by the SIGNOR resource website or downloaded for local analysis, in compliance with the FAIR principles (*Wilkinson et al., 2016*).

Second, we used the xMAP technology to interrogate signaling in FLT3-ITD cells treated with a panel of nine different perturbations. Analysis of this large multiplex signaling dataset, consisting of 16 distinct experimental conditions, revealed a clear separation between TKI-treated and untreated cells as well as IGF1 and TNFα-stimulated cells. Surprisingly, there was no clear separation between FLT3[ITD-JMD] and FLT3[ITD-TKD] cells, upon clustering based on signaling parameters. This may be caused by the targeted nature of our measurements, in line with our recent demonstration that unbiased phosphoproteome profiles discriminate FLT3-ITD cells according to the ITD location (JMD region vs TKD region). We also speculate that the different ITD insertion site has a less pronounced effect on cell signaling as compared to the pharmacological inhibition of key kinases (e.g. FLT3) or stimulation with cytokines. This observation highlights the necessity of a systems-based approach allowing the generation of predictive, genotype-specific models describing how signaling rewiring may affect TKI sensitivity.

Third, we optimized two genotype-specific Boolean models to delineate the signaling networks downstream of FLT3[ITD-JMD] and FLT3[ITD-TKD]. The topology of the two Boolean models was different and most of the interactions cell-specific, suggesting a deep rewiring of the signal downstream of FLT3 due to the different locations of the ITD. Remarkably, when we simulated the pharmacological suppression of FLT3 in silico, our models were able to recapitulate the well-documented differential sensitivity to TKI treatment of cells expressing FLT3[ITD-JMD] versus FLT3[ITD-TKD]. Additionally, by taking

advantage of two independent publicly available datasets, including phosphoproteomic and drug sensitivity screening datasets, we validated the predictions of our models.

Simulation of several simultaneous perturbations of these models in silico highlighted the role of JNK in the regulation of TKI sensitivity. Remarkably, we discovered that JNK impacts the cell cycle architecture of FLT3$^{ITD-TKD}$ cells, by acting as a mediator of the CDK1 activity. This is in line with our previously described model, showing that hitting cell cycle regulators triggers apoptosis of FLT3$^{ITD-TKD}$ cells (*Massacci et al., 2023*).

In the present study, we also investigated the clinical relevance of our optimized Boolean models, in a pilot cohort of patients. By integrating the mutation and transcriptome profiling of 14 FLT3-ITD AML patients with our cell-derived logic models, we were able to derive patient-specific signaling features and enable the identification of potential-tailored treatments restoring TKI resistance. To note, in our pilot analysis, we could observe that while our predictions were confirmed by follow-up clinical data for some patients (JMD1, JMD2, JMD3, JMD6, JMD7, JMD_TK2, JMD_TKD3, TKD2), the high genetic complexity of other FLT3-ITD positive patients was not completely addressed by our cell line-derived scaffold model (JMD5, TKD1). This could be due to a number of factors: (i) the size of the FLT3-ITD patient subgroups may have been too small to derive significant biological conclusions (e.g. only two patients with FLT3$^{ITD-TKD}$); (ii) the panel of molecular readouts in our training dataset might be too limited to capture the pleiotropic impact of the FLT3-ITD mutations; and (iii) a more heterogeneous experimental data might be needed to train a predictive model able to recapitulate the genetic background of a real cohort of patients.

In conclusion, the integration of a cell-based multiparametric dataset with a prior knowledge network in the framework of the Boolean formalism enabled us to generate optimized mechanistic models of FLT3-ITD resistance in AML. This is the proof of concept that our personalized informatics approach described here is clinically valid and will enable us to propose novel patient-centered targeted drug solutions. In principle, the generalization of our strategy will enable us to obtain a systemic perspective of signaling rewiring in different cancer types, driving novel personalized approaches.

## Materials and methods

### Cell culture

Murine Ba/F3 cells stabling expressing ITD-JMD and ITD-TKD constructs were kindly provided by Prof. T Fischer. Cells were cultured in RPMI 1640 medium (Hyclone, Thermo Scientific, Waltham, MA, USA) supplemented with 10% heat-inactivated fetal bovine serum (ECS0090D Euroclone, Italy, MI), 100 U/ml penicillin and 100 mg/ml streptomycin (Gibco 15140122), 1 mM sodium pyruvate (Sigma-Aldrich, St. Louis, MO, USA, S8636) and 10 mM 4-(2-hydroxyethyl)-1-piperazineethanesulfonic acid (HEPES) (Sigma H0887). These cells were chosen as an experimental system as previously described (*Massacci et al., 2023*).

### Multiparametric experiment of signaling perturbation

Ba/F3 FLT3$^{ITD-JMD}$ and FLT3$^{ITD-TKD}$ cells were cultured in complemented RPMI w/o FBS for 16 hr. Afterward, cells were treated with a panel of small molecule inhibitors for 90 min: Midostaurin 100 nM (Selleck Chemicals, S8064), SB203580 10 μM (Selleck Chemicals, S1076), SP600125 10 μM (Selleck

**Table 1.** Small molecule inhibitors and stimuli for the multiparametric analysis.

| Inhibitors | Target | Usage | Time | Stimuli | Usage | Time |
|---|---|---|---|---|---|---|
| Midostaurin | FLT3 | 100 nM | 90 min | IGF1 | 100 ng/ml | 10 min |
| SB203580 | p38 | 10 μM | 90 min | TNFα | 10 ng/ml | 10 min |
| SP600125 | JNK | 20 μM | 90 min | | | |
| Wortmannin | PI3K | 50 nM | 90 min | | | |
| Rapamycin | mTOR | 100 nM | 90 min | | | |
| UO126 | MEK1/2 | 15 μM | 90 min | | | |
| LY2090314 | GSK3 | 20 nM | 90 min | | | |

Chemicals, S1460), Wortmannin 35 nM (Selleck Chemicals, S2758), Rapamycin 100 nM (Sigma-Aldrich, R8781), UO126 15 μM (Sigma-Aldrich, 662005), LY2090314 20 nM (Selleck Chemicals, S7063). Cells were stimulated with IGF1 100 ng/ml (Sigma-Aldrich, I8779) and TNFα 10 ng/ml (Miltenyi Biotec, 130-101-687). *Table 1* summarizes the treatments used, inhibitors and stimuli, their specific targets, the readout sentinels, and the concentrations and the treatment time chosen. We selected the inhibitors for their specificity toward key kinases in the FLT3-ITD downstream signaling. We tested their efficacy in our model cell line to set the optimal concentration and time to inhibit the kinase activity to phosphorylate its downstream targets (i.e. the UO126 at 15 μM for 90 min inhibits MEK and we observed dephosphorylated ERK).

We selected IGF1 and TNFα as stimuli to fully activate the receptors and their downstream kinases in order to perturb and measure more efficiently the signaling. Each treatment and perturbed kinase were paired with a sentinel analyte to monitor the responses to perturbations of all main signal transduction pathways of our cell lines included in the PKN.

We combined the treatments listed in *Table 1* to finally obtain 16 different experimental conditions in both FLT3^ITD-JMD and FLT3^ITD-TKD cell lines. The experimental conditions are summarized in *Supplementary file 2* and listed below:

1. FLT3i
2. FLT3i+IGF1
3. FLT3i+TNFα
4. FLT3i+p38i+TNFα
5. FLT3i+JNKi+TNFα
6. FLT3i+PI3Ki+IGF1
7. FLT3i+mTORi+IGF1
8. FLT3i+MEKi+IGF1
9. IGF1
10. TNFα
11. p38i+TNFα
12. JNKi+TNFα
13. PI3Ki+IGF1
14. mTORi+IGF1
15. MEKi+IGF1
16. GSK3i+IGF1

**Table 2.** xMAP analytes.

| Analytes | Cat. no. | Phosphosite measured | Activity annotation |
|---|---|---|---|
| CREB1 | 42-680MAG | Ser133 | 1 |
| ERK1/2 | 42-680MAG | Thr185/Tyr187 | 1 |
| JNK | 42-680MAG | Thr183/Tyr 185 | 1 |
| p38 | 42-680MAG | Thr180/Tyr182 | 1 |
| STAT3 | 42-680MAG | Ser727 | 1 |
| STAT5 | 42-680MAG | Tyr694/699 | 1 |
| p70S6K | 42-611MAG | Thr412 | 1 |
| RPS6 | 42-611MAG | Ser235/236 | 1 |
| MTOR | 42-611MAG | Ser2448 | 1 |
| IGF1R | 42-611MAG | Tyr1135/Tyr1136 | 1 |
| PTEN | 42-611MAG | Ser380 | −1 |
| TSC2 | 42-611MAG | Ser939 | −1 |
| GSK3A | 42-611MAG | Ser21 | −1 |
| GSK3B | 42-611MAG | Ser9 | −1 |
| β-Tubulin | 46-413MAG | Total protein | Loading control |

We combined the inhibition of a specific target with the stimulation of corresponding pathways with either IGF1 (AKT-MAPK pathway) or TNFα (p38-JNK pathway), in the presence or absence of FLT3 inhibitor Midostaurin. Combinatorial treatments aimed at perturbing the cell signaling and at measuring amplified signaling changes in our system. We therefore measured the phosphorylation levels of 14 sentinel proteins listed in *Table 2* through the X-Map Luminex technology. To each residue measure, we mapped the functional role associated, activatory = 1, inhibitory = –1 depending on the annotated function on PhosphoSitePlus.

Cells were collected, lysed, and stained following the manufacturer's instructions. Briefly, in 96-well plates, cell lysates were marked with the mix of specific antibodies covalently bound to magnetic beads, and the signal was amplified with a biotin-streptavidin system. The plates were read through the Magpix instrument: for each sample, the instrument measured the intensity of the fluorescent signal pairing it with the identity of the beads given by their location on the magnetic field. As the final output, we obtained the median fluorescence intensity (MFI) for all the sentinels in each experimental condition, paired with the number of detected beads. For each sentinel, the fluorescent threshold should be associated with a count of more than 50 beads to be technically reliable. We then excluded from the dataset the measures with less than 50 beads detected as shown in the 'filter on n. beads' sheet of *Supplementary file 3* and *Supplementary file 4*. Then, we normalized the MFI of each analyte on the values of β-tubulin as loading control and we calculated the median and SD of the three biological replicates (*Supplementary file 3* and *Supplementary file 4*).

## Data normalization

The phosphorylation measure of the 14 sentinel signaling proteins was scaled between 0 and 1, using customized Hill functions for each analyte. By applying the formula:

$$y = \frac{x^n}{K + x^n}$$

We derived n and K parameters of customized Hill functions from the distribution of each analyte in our experimental data. Briefly, given the asymptotic behavior of the Hill function, we set the experimental maximum (maxS) of the analyte to be 0.999 (theoretical maximum, maxT) and the experimental minimum (minS) to be 0.001 (theoretical minimum, minT). We then computed the b parameter as:

$$b = \frac{\max S}{\min S}$$

Next, we calculated n and K for each analyte Hill function:

$$n = \frac{(1 - \min T)\,\max T}{(1 - \max T)\,\min T}$$

$$K = \frac{1 - \max T}{\max T}\max S^n$$

## PCA and hierarchical clustering

PCA was performed using the *stats* R package (v. 4.1.2).

Perseus software was employed to perform unsupervised hierarchical clustering. Specifically, the Pearson correlation coefficents between phosphorylation profiles of sentinel proteins across different experimental conditions were calculated and used to generate the tree. Similar experimental conditions are in the same branches.

## FLT3-ITD-specific Boolean model construction with CellNetOptimizer

We exploited the CellNetOptimizer pipeline which integrates (i) a PKN and (ii) multiparametric, normalized experimental data to obtain two FLT3 TD-JMD and -TKD dynamic and predictive Boolean models.

The overall bioinformatics strategy was divided into two main parts:

A. Cells-specific models' generation
B. Assessment and validation of the optimized models

For each step, we provide references to the corresponding code available in our GitHub repository, (copy archived at *SaccoPerfettoLab, 2023*). This meticulous exposition serves to enhance transparency and facilitate a more in-depth and independent assessment of our analytical procedures.

## Cell-specific models' generation

To generate FLT3-ITD-specific predictive Boolean models, we exploited the genetic algorithm of Cell-NOptR, which trains a manually curated FLT3 Prior Knowledge Network (76 nodes and 193 edges, *Figure 2—figure supplement 1*) on perturbation data (*Supplementary file 1*).

### PKN manual curation

We built an FLT3-ITD-specific PKN combining (i) a manual curation and (ii) a data-driven approach. Starting from the SIGNOR database, through a curation effort, we assembled a causal network describing the FLT3-ITD signaling, comprising all the direct and indirect interactions implied in the receptor signaling and leukemogenesis. The PKN is publicly available on SIGNOR. We downloaded from SIGNOR the interaction table representing the PKN, and we manually simplified the network, we compressed some articulated and redundant paths (*Supplementary file 1*; *Figure 2—figure supplement 2*). We converted the network into a .sif file made of three columns, entity A, entity B, and interaction type described with 1 if activatory or –1 if inhibitory. Importantly, during the optimization process, the PKN was adjusted until we reached an optimal performance of the model. The final version of the PKN displayed 76 nodes and 193 edges.

### Preprocessing of the PKN

Next, we used CellNOptR v.1.40.0, to preprocess the PKN and to convert the causal network into logical functions (scaffold model), describing the regulatory relations among gene products using OR/AND Boolean operators. This phase is needed to achieve, from the PKN, a simplified yet informative network (here dubbed 'scaffold model'). We remove unnecessary nodes such as chains of signaling interaction with the same sign, and we check that the scaffold model has the potential to display an output (modulation of a sentinel) for any input provided (cue) (*Terfve et al., 2012*). This phase accounts for three main steps:

- *Compression*: unmeasured and untargeted proteins are removed.
- *Expansion*: the remaining nodes are connected to every possible combination of upstream regulators with both 'AND' and 'OR' Boolean operators. The 'AND' operators are added as nodes of the network. Thus, we end with 204 nodes (30 proteins and 174 AND Boolean operators) and 612 edges.
- *Imputation*: This step is crucial to add co-regulations that may happen in our system but are still not known or annotated in literature and thus are missing in the PKN. The R package *CNOR-feeder* v.1.34.0 (FEED method) was exploited to integrate the interactions derived from correlation among analytes in the perturbation data. This step allowed us to add 144 data-derived edges in the FLT3$^{ITD-JMD}$ model and 170 in the FLT3$^{ITD-TKD}$.

Using this strategy, we obtained two FLT3-ITD-specific scaffold models, accounting for 206 and 208 nodes and 756 and 782 edges, for FLT3$^{ITD-JMD}$ and FLT3$^{ITD-TKD}$, respectively.

The code to reproduce this step is available in our GitHub repository in the 'Cell-specific model's generation' section, Paragraph 2.

### Network model optimization

In this step, we took advantage of a successful strategy used in our previous work (*Sacco et al., 2012*). The aim of this step is to train and optimize a Boolean logic model able to reproduce in silico our cue-sentinel-response multiparametric training dataset (*Figure 3A*). In this phase, we followed standard practice in logical modeling (*Dorier et al., 2016*; *Traynard et al., 2017*):

- *Generation of the family of 1000 Boolean models*: we run the genetic algorithm of CellNOptR package 1000 times to generate a family of 1000 optimized Boolean models for each cell line. Briefly, this choice is driven by:
  - the stochastic nature of the genetic algorithm.
  - the possibility of obtaining quantitative predictions averaging the discrete node state in each model. This feature offers the opportunity to readily evaluate the optimization process,

since the agreement between experimental and simulated data increases when the predictions are carried out by averaging a larger number of models.

The CellNOptR genetic algorithm minimizes the difference (mean squared error [mse]) between experimental data and the values simulated from the Boolean model.

- *Selection of 100 optimal models*: given the observation that the agreement (mse) between experimental and simulated data reached an apparent plateau at 100 models (*Figure 3—figure supplement 1*), we filtered out only the 100 models that better fit the experimental data, thus having the lowest mse (family of best models) (*Dorier et al., 2016*). Then, for each cell line, we compared the experimental activity modulation of each protein (*Figure 3A*, panel 1) with the average activity modulation of the protein in the family of 100 models (*Figure 3A*, panel 2). The fit between simulated and experimental data is reported in *Figure 3A*, panel 3. Importantly, this procedure enables quantitative prediction even using Boolean models, which are discrete by nature.

The code to reproduce this step is available in our GitHub repository in the 'Cell-specific model's generation' section, Paragraph 3.

## Visualization of optimized models

- *Final model selection*: to perform further analyses we selected only the best model. The strength of this approach is that Boolean rules are derived from the combination of prior knowledge and experimental data. However, the optimization process creates a family of Boolean models where some nodes are regulated by different and opposite Boolean rules making it difficult to perform simulations (*Figure 3—figure supplement 1B*). To resolve this ambiguity and have a single Boolean rule for each node and, hence, to exploit the intrinsic predictive power of Boolean models, we selected the model with the lowest error (mse) between experimental and simulated data in FLT3$^{ITD-JMD}$ and FLT3$^{ITD-TKD}$ cell lines (best model). The so-obtained final models consist of 68 and 60 nodes (of which 38 and 30 are AND operators) and 161 and 133 edges for FLT3$^{ITD-JMD}$ and FLT3$^{ITD-TKD}$, respectively.
- *High-confidence model generation*: to keep a measure of reliability from the whole optimization procedure in each best model, we added the frequency of each edge in the 100 models as an attribute in each cell-specific best model. The two final Boolean models with the highest edge confidence (frequency in 100 models >0.4) are shown in *Figure 3B and C*.

The code to reproduce these steps is available in our GitHub repository in the 'Visualization of optimized models' section.

## Assessment and validation of the optimized models
### Validation strategy

To assess whether the newly generated FLT3$^{ITD-JMD}$ and FLT3$^{ITD-TKD}$ Boolean models could recapitulate in silico the TKI-induced modulation of apoptosis and proliferation, we defined a strategy that is composed of three steps: (i) We set up two initial conditions of the models aiming at reproducing the biological context of our cells before and after the FLT3 inhibitor treatment. (ii) We computed the steady state of each Boolean model to derive the states of the model's proteins associated with each initial condition. (3) We unbiasedly associated the final conditions with the phenotypic states of the cells.

We here in brief describe the rationale behind these choices and relative technical details.

1. *Initial conditions of the simulation*: briefly, we defined '*Untreated*' the condition that reproduces a malignant state where cells proliferate and escape apoptosis. From the model perspective, this condition is represented by an active state of the FLT3, IGF1R, and TNFR nodes. Next, we defined '*FLT3i*' the condition that reproduces malignant cells treated with the FLT3 inhibitor (Midostaurin). From the model perspective, this condition is represented by an active state of the IGF1R, and TNFR nodes and an inactive state of FLT3. Summing up, the conditions used were:
   - *Untreated condition,* in which all the receptors included in the model (FLT3, IGF1R, and TNFR) are set to ON.
   - *FLT3 inhibition condition (FLT3i),* in which FLT3 was set to OFF.

2. *Computation of steady states*: here we used the *simulatorT1* function of CellOptR package, which performs a Synchronous Boolean simulation to compute the steady state of each cellular model in two conditions (*Figure 3—figure supplement 1D*).

3. *Apoptosis and proliferation activity inference*: to functionally interpret the results of the simulations, we derived the levels of 'apoptosis inhibition' and 'proliferation activation' in each condition (*Figure 3E*). To do that:

   - We annotated all the proteins in the network as activators or inhibitors of 'apoptosis' and 'proliferation' phenotypes using our recently published research ProxPath (*Lo Surdo et al., 2023*), which computes significantly 'close' paths linking SIGNOR proteins and phenotypes. The distance table connecting the model nodes to the 'apoptosis' and 'proliferation' phenotypes is available in *Supplementary file 1*.
   - For the inference of phenotypes, we considered only proteins that were (i) regulators of a phenotype and (ii) endpoint proteins in high-confidence signaling axes (edge frequency 0.4, *Figure 3B and C*). This choice is driven by the need to avoid redundancy and consider the final nodes of the model which are the real effector proteins according to our mutations-specific FLT3 Boolean models.
   - Then, we integrated the signal of the phenotype regulators proteins (*Figure 3E* heatmaps) to compute the level of 'apoptosis inhibition' and 'proliferation activation' in each cell line (*Figure 3E* bar plot). Integration, in this context, means computing the sum of the 'scores' of proteins that influence each phenotype, with an underlying assumption of equal importance for both inhibitors and activators (OR logic).

As a result of this step, we obtained a comprehensive model encompassing proteins and phenotypes (apoptosis and proliferation) that can possess multiple values, effectively creating a multi-valued model. This enhancement enables a finer comparison between different cell lines. The code that describes this step is available in our GitHub repository in the 'In silico validation' section.

## Boolean models' validation using independent resources

Using the strategy described above, we computed the steady state of FLT3[ITD-JMD] and [ITD-TKD] Boolean models with and without the inhibition of FLT3 and other druggable nodes.

To independently validate the models, we used as a reference the phosphoproteomic data of FLT3-ITD Ba/F3 cells (FLT3[ITD-JMD] and FLT3[ITD-TKD]) upon quizartinib (AC220) treatment (*Massacci et al., 2023*). We mapped xMAP residues associated with protein complexes (e.g. ERK1/2) to unique protein sequences (e.g. Mapk1 and Mapk3) (*Supplementary file 2*). We estimated the activity of sentinel proteins in the reference dataset using the modulation of their regulatory phosphosites. Then, we compared the estimated activity with the sentinels' states in the FLT3 inhibition simulation (*Figure 3—figure supplement 2*).

Moreover, to functionally interpret the results and assess the reliability of the model, we computed the activity of 'apoptosis' and 'proliferation' phenotypes upon FLT3 and other druggable nodes' inhibition as described in *3. Apoptosis and proliferation activity inference* section.

The Beat AML program on a cohort of 672 tumor specimens collected from 562 patients has been exploited for model validation. We focused on drug sensitivity screening on 134 patients carrying the typical FLT3-ITD mutation in the JMD region. Drugs were annotated for their targets using SIGNOR and ChEBI databases. Drugs inhibiting FLT3, PI3K, mTOR, JNK, and p38 were selected and the average IC50 of FLT3[ITD-JMD] patient-derived primary blasts was calculated (*Supplementary file 6*).

## Combinatory treatment inference

Given the ability of the FLT3[ITD-JMD] and FLT3[ITD-TKD] networks to reproduce the different sensitivity to TKIs, the generated ITD-specific Boolean models were exploited to find a co-treatment, in addition to FLT3 inhibition, that could revert the resistant phenotype.

For each cell line, we inferred the levels of 'apoptosis inhibition' and 'proliferation activation' upon every possible combined inhibition of FLT3 and key signaling kinases (ERK1/2, MEK1/2, GSK3A/B, IGF1R, JNK, KRAS, MEK1/2, MTOR, PDPK1, PI3K) in the model (*Figure 4A*). Briefly, here we used the same strategy described in 'Assessment and validation of the optimized models' section. The initial conditions considered were:

- *Untreated*, in which all the receptors included in the model (FLT3, IGF1R, and TNFR) are set to ON.
- *FLT3i*, in which FLT3 was set to OFF.

- *Combinatory treatment* (e.g. FLT3i+KRASi), in which both FLT3 and the target of the specific inhibition (in this case KRAS) were set to OFF.

We eventually selected co-treatments in the FLT3$^{ITD-TKD}$ model able to trigger activation levels of the 'apoptosis' and 'proliferation' to the same level as the FLT3$^{ITD-JMD}$ model.

The code to reproduce this step is available in our GitHub repository in the 'Combinatory treatment inference' section.

## Apoptosis assay

Ba/F3 cells were treated with Midostaurin 100 nM and SP600125 10 µM for 24 hr.

The concentration of SP600125 to use for this long-term treatment was chosen based on setup experiment: we treated sensitive and resistant cells with increasing concentrations of SP600125 for 24 hr and evaluated the cell viability using the Cell Proliferation Kit I (MTT) (Roche, Cat. 11465007001) and measuring the absorbance value at $\lambda$ =590 nm. We then calculated the IC50 with a nonlinear regression drug-response curve fit using Prism 7 (GraphPad). IC50 values are approximately 1.5 µM in FLT3-ITD mutant cell lines (FLT3$^{ITD-JMD}$ cells IC50=1.54 µM; FLT3$^{ITD-TKD}$ cells IC50=1.69 µM). The SP600125 treatment affects cell viability, reaching a plateau phase of cell death and at about 2 µM. We used 10 µM SP600125 to inhibit JNK phosphorylation (*Kim et al., 2010*; *Moon et al., 2009*). The concentration of Midostaurin for the apoptotic assay was chosen based on the previously published work (*Massacci et al., 2023*) where we show that FLT3$^{ITD-TKD}$ cells treated with Midostaurin 100 nM show lower apoptotic rate and higher cell viability compared to FLT3$^{ITD-JMD}$ cells. Apoptotic cells were analyzed with the Ebioscience Annexin V Apoptosis Detection Kit APC according to the kit instructions (Cat. 88-8007-74, Thermo Fisher Scientific). Samples were read through the CytoFLEX S (Beckman Coulter) instrument using the APC laser to detect the Annexin-V+ cells. Quality control of the cytometer was assessed weekly using CytoFLEX Daily QC Fluorospheres (Beckman Coulter B53230). The fluorescence threshold was set for the APC laser using a blank sample, without the fluorescent label. The results were analyzed by the CytExpert software and represented in bar plots as the percentage of Annexin-V+ cells fold change of treated conditions on controls.

## MTT assays

Ba/F3 cells were treated with Midostaurin 100 nM, SP600125 20 µM, SB203580 10 µM, and UO126 15 µM for 24 hr in 96-well plates. The concentration of SB203580 and UO126 to use for this long-term treatment was chosen based on previous data available in the lab and setup experiments. We used the concentration in which we could observe a reduced activity of the target (lower phosphorylation of downstream proteins) without cell toxicity. Cell viability was assessed using the Cell Proliferation Kit I (MTT) (Roche, Cat. 11465007001), following the manufacturer's instructions. The plates were read through a microplate reader (Bio-Rad) at $\lambda$ =590 nm. The results were represented in bar plots as fold change of treated conditions on controls.

## Cell cycle analysis

Ba/F3 cells were treated with Midostaurin 100 nM and SP600125 10 µM for 24 hr, 106 cells were collected, washed in ice-cold PBS 1×, and fixed in agitation with 70% cold ethanol. Fixed samples were incubated at 4°C O/N, washed in PBS 1×, and resuspended in 1 µg/ml DAPI (Thermo Scientific, #62248) and 0.2 mg/ml RNase (Thermo Scientific, # 12091021) PBS solution before analysis. Samples were read through the CytoFLEX S (Beckman Coulter) instrument using the PB450 laser to detect the DAPI+ cells. Data were analyzed by CytExpert (Beckman Coulter) software and represented in bar plots as a percentage of DAPI+ cells.

## Immunoblot analysis

Ba/F3 cells were seeded at the concentration of 5×105 cells/ml and treated with Midostaurin 100 nM, SP600125 20 µM, SB203580 10 µM, UO126 15 µM, for 24 hr, and TNFα 10 ng/ml for 10 min. Cells were lysed for 30 min in ice-cold lysis buffer (150 mM NaCl, 50 mM Tris-HCl pH 7.5, 1% Nonidet P-40 [NP-40], 1 mM EGTA,5 mM MgCl$_2$, 0.1% SDS) supplemented with 1 mM PMSF, 1 mM orthovanadate, 1 mM NaF, protease inhibitor mixture 1×, inhibitor phosphatase mixture II 1×, and inhibitor phosphatase mixture III 1×. The insoluble material was separated at 13,000 rpm for 30 min at 4°C and total protein concentration was measured on supernatants using Bradford reagent (Bio-Rad).

Protein extracts were denatured with NuPAGE LDS (Invitrogen) and boiled at 95°C for 10 min. SDS-PAGE and transfer were performed on 4–15% Bio-Rad Mini/Midi PROTEAN TGX and Trans-Blot Turbo mini/midi nitrocellulose membranes using the Trans-Blot Turbo Transfer System (Bio-Rad). Nonspecific binding sites were blocked using 5% non-fat dried milk in TBS-0.1% Tween-20 (TBS-T) for 1 hr at RT, shaking. Primary antibodies were diluted according to the manufacturer's instruction and incubated at 4°C O/N, shaking. HRP-conjugated secondary antibodies were diluted 1:3000 in 5% non-fat dried milk in TBS-T and incubated for 1 hr at RT, shaking. Peroxidase chemiluminescence reaction was enhanced with Clarity Western ECL Blotting Substrates (Bio-Rad) and detected through the Chemidoc detection system (Bio-Rad). Densitometric quantitation of raw images was obtained with Fiji software (ImageJ, NIH). The primary and secondary antibodies used are listed: CDK1 (sc- 53219); CDK2 (sc-6248); phpspho-CDK1 Y15 (CS 9111); phospho-CDK1 T161 (CS 9114); phospho-CDK2 T160 (CS 2561); phospho-SAPK/JNK T183/Y185 (CS 9251); SAPK/JNK (CS 9252); CyclinB1 (CS 4138); CyclinE2 (CS 4132); Vinculin (CS 13901).

## Primary patient blast sensitivity

Peripheral blood (PB) samples were collected from patients with AML in accordance with the Declaration of Helsinki (ethics committee approval number 115/08) and with the patients' informed consent. The FLT3-ITD mutation integration site was determined as previously described (*Massacci et al., 2023*).

Mononuclear cells were isolated from PB using Ficoll-Paque (GE Healthcare, Chicago, IL, USA). Cryoconserved peripheral blood mononuclear cells (PBMCs) from seven patients were cultured in RPMI-1640 medium (Sigma-Aldrich, St. Louis, MO, USA) supplemented with 10% fetal calf serum (Bio&Sell GmbH, Germany), 2 mM L-glutamine (Sigma-Aldrich), and 40 U/ml penicillin-streptomycin (Thermo Fisher Scientific) at a density of $5 \times 10^5$/ml. Cultures were incubated for 48 hr in the absence or presence of 100 nM PKC412 and 10 µM SP600125 (all Selleck Chemicals LLC, Houston, TX, USA), or combinations of PKC412 with SP600125. The viability of the patient's blast cells was assessed by flow cytometry, while the specific cell death was calculated as described previously (*Pugliese et al., 2023*). Briefly, samples were stained with fluorochrome-conjugated antibodies against CD45, CD33, CD34, CD13, and CD117, followed by the addition of Annexin V and 7AAD. The samples were recorded using a NorthernLight-3000 spectral flow cytometer (Cytek Biosciences, Freemont, CA, USA) and analyzed through FlowJo v.10.9 (BD Bioscience, Franklin Lakes, NJ, USA).

## Statistical analysis

Data are represented as the mean ± SEM of at least three independent experimental samples (n=3). Comparisons between three or more groups were performed using the ANOVA test. Statistical significance between the two groups was estimated using Student's t-test. Statistical significance is defined as p-value where *$p<0.05$; **$p<0.01$; ***$p<0.001$; ****$p<0.0001$. All statistical analyses were performed using Prism 7 (GraphPad).

## RNAseq of patient samples

RNA was extracted from peripheral blasts of 14 treatment naïve patients with de novo FLT3-ITD-driven AML diagnosis. Clinical characteristics were available for only 10 patients (*Supplementary file 9*). PB samples from 14 AML patients were obtained upon the patient's informed consent. The integration site of the FLT3-ITD mutation was determined as previously described (*Rücker et al., 2022*). Briefly, RNA was prepared from PBMCs using the RNeasy Mini Kit (QIAGEN, Germany), retro-transcribed in cDNA, and quantified using the Qubit 4.0 Fluorimeter assay (Thermo Fisher Scientific) and sample integrity, based on the DIN (DNA integrity number), was assessed using a Genomic DNA ScreenTape assay on TapeStation 4200 (Agilent Technologies).

High-throughput sequencing was performed on the coding region of 262 genes involved in hematologic malignancies. A comprehensive list of all genes is included in *Supplementary file 7*.

Genomic DNA Libraries were prepared from 100 ng of total DNA using the NEGEDIA Cancer Haemo Exome sequencing service (Next Generation Diagnostic srl) which included library preparation, target enrichment using Cancer Haemo probe set, quality assessment, and sequencing on a NovaSeq 6000 sequencing system using a paired-end, 2×150 cycle strategy (Illumina Inc). Paired-end reads were produced with a 100× coverage and a median of 40M reads per sample.

## Bioinformatic analysis of transcriptome data of patient samples

The raw data were analyzed by Next Generation Diagnostics srl Cancer haemo Exome pipeline (v.1.0) which involves a cleaning step by UMI removal, quality filtering and trimming, alignment to the reference genome, removal of duplicate reads, and variant calling (FASTQC: https://www.bioinformatics.babraham.ac.uk/projects/fastqc/, *Freed et al., 2017*; *McLaren et al., 2016*; *Smith et al., 2017*).

Illumina NovaSeq 6000 base call (BCL) files were converted into fastq files through bcl2fastq (2019) (v.2.20.0.422). UMI removal was carried out with UMI-tools 1.1.1. Data quality control was performed with FastQC v.0.11.9 and reads were trimmed and cleaned using Trimmomatic 0.38.

Alignment to human reference (hg38, GCA_000001405.15), deduplication, and variant calling were performed with Sentieon 202011.01.

Variant calling output was converted into vcf and MAF format. The maf files were further annotated with the OncoKB annotator (*Chakravarty et al., 2016*) to obtain the mutation effect on protein function and oncogenicity. We filtered out variants with MUTATION_EFFECT equal to 'Unknown', 'Inconclusive', 'Likely Neutral', and 'Switch-of-function' (*Supplementary file 9*).

All downstream analyses were carried out using R v.4.1.2 and BioConductor v.3.13 (*Huber et al., 2015*; *R Development Core Team, 2021*).

Gene counts tables were generated from bam files using Rsubread v. 2.8.2. Data were normalized with the 'trimmed mean of M values' (TMM) method of edgeR v.3.36.0 (*Robinson et al., 2010*) and converted to log2 (*Supplementary file 9*).

## FLT3-ITD localization

Generic variant callers can't identify medium-sized insertions, like FLT3-ITDs. As such, we used the specialized algorithm getITD v.1.5.16 (*Blätte et al., 2019*) to localize ITDs in each patient. Reads mapping on FLT3 genomic region between JMD and TKD domain (28033888–28034214) were extracted from bam files and converted in fastq format with samtools v.1.16.1.

getITD was run with default parameters, but with a custom reference sequence without introns ('caat ttaggtatgaaagccagctacagatggtacaggtgaccggctcctcagataatgagtacttctacgttgatttcagagaatatgaata tgatctcaaatgggagtttccaagagaaaatttagagtttgggaaggtactaggatcaggtgctttttggaaaagtgatgaacgcaaca gcttatggaattagcaaaacaggagtctcaatccaggttgccgtcaaaatgctgaaag') that was annotated according to getITD annotation file. Patients carried multiples ITDs and we classified them as follows: eight patients as FLT3^ITD-JMD^ (JMD1-8), four patients as FLT3^ITD-JMD+TKD^ (JMD_TKD1-4), and two patients as FLT3^ITD-TKD^ (TKD1-2) (*Supplementary file 8*).

## Patients' simulation on cell-derived ITD-specific Boolean models

We decided to use the FLT3^ITD-JMD^ cell models for both FLT3^ITD-JMD^ and FLT3^ITD-JMD+TKD^ given the dominant effect that the FLT3^ITD-JMD^ mutation displays over the FLT3^ITD-TKD^ one (*Rücker et al., 2022*).

We used patients' data to perform Boolean simulations on cell-derived ITD-specific Boolean models. We used FLT3^ITD-JMD^ cell lines Boolean models for FLT3^ITD-JMD^ and FLT3^ITD-JMD+TKD^ patients and FLT3^ITD-TKD^ models for FLT3^ITD-TKD^ patients. Each patient mutational profile was binarized, setting 'Loss-of-function' or 'Likely loss-of-function' equal to 0, and 'Gain-of-function' or 'Likely gain-of-function' equal to 1. Using CellNOptR, for each patient we computed the steady state of its correspondent cell-derived Boolean model in two conditions: (i) the input is the mutational profile (tumor simulation – 'untreated'); (ii) the input is the mutational profile + FLT3 inhibition (treatment with FLT3i simulation). Hence, we inferred the 'apoptosis' and 'proliferation' phenotype scores, as stated above.

We also performed an in silico combined treatment of patients simulating their mutations with the inhibition of FLT3 and a key signaling kinase (11 different simulations).

## Network visualization

To visualize the best model, we exported CellNOptR results using writeScaffold and writeNetwork functions and imported the network in Cytoscape v.3.9.0. For nodes and edges' color, we used the style of Cytocopter app v.3.9. We added as edges' attribute the frequency of each edge in the family of 100 models and we kept only the edges having a frequency >0.4.

## Statistical analyses and tools

The patient z-score was computed by subtracting the mean of patient-specific expression distribution and dividing by its standard deviation. PCA and clustering graphs were generated with stats v.4.1.2, ggfortify v.0.4.15, and ggdendrogram v.0.1.23 packages. Heatmaps were created using pheatmap v.1.0.12. Phenotypes bar plots are created with ggplot2 v.3.4.0.

## Acknowledgements

We thank Prof. Gianni Cesareni and Prof. Luisa Castagnoli for their essential scientific input. Dr. Serena Paoluzi for their technical support. This research was funded by the Italian Association for Cancer Research (AIRC) with a grant to FS, GMP, GM, and VB (Start-Up grant number 21815) and LP (MFAG 2023 - ID. 28858 project). SL, VV, and SG are supported by MUR.

## Additional information

### Funding

| Funder | Grant reference number | Author |
| --- | --- | --- |
| Fondazione AIRC per la ricerca sul cancro ETS | 21815 | Francesca Sacco Giusj Monia Pugliese Giorgia Massacci Valeria Bica |
| Fondazione AIRC per la ricerca sul cancro ETS | 28858 | Livia Perfetto |
| Italian Minister of University (MUR) | | Sara Latini Veronica Venafra Simone Graziosi |

The funders had no role in study design, data collection and interpretation, or the decision to submit the work for publication.

### Author contributions

Sara Latini, Conceptualization, Data curation, Formal analysis, Validation, Investigation, Visualization, Methodology, Writing – original draft, Writing – review and editing; Veronica Venafra, Conceptualization, Data curation, Software, Formal analysis, Validation, Investigation, Visualization, Methodology, Writing – original draft, Writing – review and editing; Giorgia Massacci, Valeria Bica, Simone Graziosi, Giusj Monia Pugliese, Gerardo Pepe, Formal analysis, Methodology, Writing – review and editing; Marta Iannuccelli, Data curation, Formal analysis, Writing – review and editing; Filippo Frioni, Gessica Minnella, Resources, Formal analysis, Writing – review and editing; John Donald Marra, Formal analysis, Writing – review and editing; Patrizia Chiusolo, Dimitros Mougiakakos, Martin Böttcher, Thomas Fischer, Resources, Formal analysis, Investigation, Writing – review and editing; Manuela Helmer Citterich, Formal analysis, Investigation, Writing – review and editing; Livia Perfetto, Conceptualization, Data curation, Software, Formal analysis, Supervision, Funding acquisition, Validation, Investigation, Visualization, Methodology, Writing – original draft, Project administration, Writing – review and editing; Francesca Sacco, Conceptualization, Resources, Data curation, Software, Formal analysis, Supervision, Funding acquisition, Validation, Investigation, Visualization, Methodology, Writing – original draft, Project administration, Writing – review and editing

### Author ORCIDs

Sara Latini http://orcid.org/0000-0002-2353-3155
Veronica Venafra http://orcid.org/0000-0003-3830-618X
Patrizia Chiusolo https://orcid.org/0000-0002-1355-1587
Martin Böttcher https://orcid.org/0000-0003-2911-8830
Francesca Sacco https://orcid.org/0000-0001-5586-9529

## Ethics

Human subjects: Peripheral blood (PB) samples were collected from patients with acute myeloid leukemia (AML) in accordance with the Declaration of Helsinki (ethics committee approval number 115/08) and with the patients' informed consent.

Reviewer #1 (Public Review): https://doi.org/10.7554/eLife.90532.3.sa1
Reviewer #2 (Public Review): https://doi.org/10.7554/eLife.90532.3.sa2
Reviewer #3 (Public Review): https://doi.org/10.7554/eLife.90532.3.sa3
None https://doi.org/10.7554/eLife.90532.3.sa4

# Additional files

## Supplementary files

• Supplementary file 1. FLT3-internal tandem duplication (ITD) prior knowledge network (PKN), table downloaded form SIGNOR, data-driven edges integration, PKN in .sif format; regulators of phenotypes annotated by ProxPath resource.

• Supplementary file 2. Experimental design of the multiparametric experiment of FLT3 [ITD-JMD] and [ITD-TKD] BaF3 cell lines; treatments and analytes measured; activity readout annotation.

• Supplementary file 3. Cue-sentinel-response multiparametric dataset raw data and statistics (MILLIPLEX kit: 9plex_Cat.No.48-680MAG).

• Supplementary file 4. Cue-sentinel-response multiparametric dataset, raw data, and statistics (MILLIPLEX kit: 11plex_Cat.No.48-611MAG).

• Supplementary file 5. Complete cue-sentinel-response multiparametric dataset used for Boolean models building in MIDAS format, raw and normalized data.

• Supplementary file 6. Data used for the validation of FLT3-internal tandem duplications (ITDs) Boolean models using independent resources.

• Supplementary file 7. Panel of 262 mutations relevant to hematological malignancies analyzed in de novo acute myeloid leukemia (AML) cohort of 14 patients.

• Supplementary file 8. Results of the getITD output for the classification of the de novo acute myeloid leukemia (AML) cohort of 14 patients.

• Supplementary file 9. Clinical data, mutation profile, and RNAseq results of the acute myeloid leukemia (AML) patients' cohort.

• MDAR checklist

## Data availability

Curated data have been submitted to SIGNOR for reuse and interoperability and can be accessed here. Transcriptomic data of patients has been submitted to GEO and can be accessed using the accession number: GSE247483. The code developed for the study has been organized on a GitHub page (copy archived at *SaccoPerfettoLab, 2023*).

The following dataset was generated:

| Author(s) | Year | Dataset title | Dataset URL | Database and Identifier |
|---|---|---|---|---|
| Latini S, Venafra V, Massacci G, Bica V, Graziosi S, Pugliese GM, Iannuccelli M, Frioni F, Minnella G, Marra JD, Chiusolo P, Pepe G, Helmer-Citterich M, Mougiakakos D, Boettcher M, Fischer T, Perfetto L, Sacco F | 2024 | Unveiling the signaling network of FLT3-ITD AML improves drug sensitivity prediction | https://www.ncbi.nlm.nih.gov/geo/query/acc.cgi?acc=GSE247483 | NCBI Gene Expression Omnibus, GSE247483 |

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
