## [Editor Report · eLife assessment]

This **important** study could potentially represent a step forward towards personalized medicine by combining cell-based data and a prior-knowledge network to derive Boolean-based predictive logic models to uncover altered protein/signaling networks within cancer cells. The level of evidence supporting the conclusions is **solid**, as the authors present analyses on independent, real-world data to validate their approach. These findings could be of interest to medical biologists working in the field of cancer, as the work should inform drug development and treatment choices in the field of oncology.

---

## [Referee Report · Reviewer #1 (Public Review)]

The authors deploy a combination of their own previously developed computational methods and databases (SIGNOR and CellNOptR) to model the FLT3 signaling landscape in AML and identify synergistic drug combinations that may overcome the resistance AML cells harboring ITD mutations in the TKI domain of FLT3 to FLT3 inhibitors. I did not closely evaluate the details of these computational models since they are outside of my area of expertise and have been previously published. The manuscript has significant issues with data interpretation and clarity, as detailed below, which, in my view, call into question the main conclusions of the paper.

The authors train the model by including perturbation data where TKI-resistant and TKI-sensitive cells are treated with various inhibitors and the activity (i.e. phosphorylation levels) of the key downstream nodes are evaluated. Specifically, in the Results section (p. 6) they state "TKIs sensitive and resistant cells were subjected to 16 experimental conditions, including TNFa and IGF1 stimulation, the presence or absence of the FLT3 inhibitor, midostaurin, and in combination with six small-molecule inhibitors targeting crucial kinases in our PKN (p38, JNK, PI3K, mTOR, MEK1/2 and GSK3)". I would appreciate more details on which specific inhibitors and concentrations were used for this experiment. More importantly, I was very puzzled by the fact that this training dataset appears to contain, among other conditions, the combination of midostaurin with JNK inhibition, i.e. the very combination of drugs that the authors later present as being predicted by their model to have a synergistic effect. Unless my interpretation of this is incorrect, it appears to be a "self-fulfilling prophecy", i.e. an inappropriate use of the same data in training and verification/test datasets.

My most significant criticism is that the proof-of-principle experiment evaluating the combination effects of midostaurin and SP600125 in FLT3-ITD-TKD cell line model does not appear to show any synergism, in my view. The authors' interpretation of the data is that the addition of SP600125 to midostaurin rescues midostaurin resistance and results in increased apoptosis and decreased viability of the midostaurin-resistant cells. Indeed, they write on p.9: "Strikingly, the combined treatment of JNK inhibitor (SP600125) and midostaurin (PKC412) significantly increased the percentage of FLT3ITD-TKD cells in apoptosis (Fig. 4D). Consistently, in these experimental conditions, we observed a significant reduction of proliferating FLT3ITD- TKD cells versus cells treated with midostaurin alone (Fig. 4E)." However, looking at Figs 4D and 4E, it appears that the effects of the midostaurin/SP600125 combination are virtually identical to SP600125 alone, and midostaurin provides no additional benefit. No p-values are provided to compare midostaurin+SP600125 to SP600125 alone but there seems to be no appreciable difference between the two by eye. In addition, the evaluation of synergism (versus additive effects) requires the use of specialized mathematical models (see for example Duarte and Vale, 2022). That said, I do not appreciate even an additive effect of midostaurin combined with SP600125 in the data presented.

In my view, there are significant issues with clarity and detail throughout the manuscript. For example, additional details and improved clarity are needed, in my view, with respect to the design and readouts of the signaling perturbation experiments (Methods, p. 15 and Fig 2B legend). For example, the Fig 2B legend states: "Schematic representation of the experimental design: FLT3 ITD-JMD and FLT3 ITD-JMD cells were cultured in starvation medium (w/o FBS) overnight and treated with selected kinase inhibitors for 90 minutes and IGF1 and TNFa for 10 minutes. Control cells are starved and treated with PKC412 for 90 minutes, while "untreated" cells are treated with IGF1 100ng/ml and TNFa 10ng/ml with PKC412 for 90 minutes.", which does not make sense to me. The "untreated" cells appear to be treated with more agents than the control cells. The logic behind cytokine stimulation is not adequately explained and it is not entirely clear to me whether the cytokines were used alone or in combination. Fig 2B is quite confusing overall, and it is not clear to me what the horizontal axis (i.e. columns of "experimental conditions", as opposed to "treatments") represents. The Method section states "Key cell signaling players were analyzed through the X-Map Luminex technology: we measured the analytes included in the MILLIPLEX assays" but the identities of the evaluated proteins are not given in the Methods. At the same time, the Results section states "TKIs sensitive and resistant cells were subjected to 16 experimental conditions" but these conditions do not appear to be listed (except in Supplementary data; and Fig 2B lists 9 conditions, not 16). In my subjective view, the manuscript would benefit from a clearer explanation and depiction of the experimental details and inhibitors used in the main text of the paper, as opposed to various Supplemental files/figures. The lack of clarity on what exactly were the experimental conditions makes the interpretation of Fig 2 very challenging. In the same vein, in the PCA analysis (Fig 2C) there seems to be no reference to the cytokine stimulation status while the authors claim that PC2 stratifies cells according to IGF1 vs TNFalpha. There are numerous other examples of incomplete or confusing legends and descriptions which, in my view, need to be addressed to make the paper more accessible.

I am not sure that I see significant value in the patient-specific logic models because they are not supported by empirical evidence. Treating primary cells from AML patients with relevant drug combinations would be a feasible and convincing way to validate the computational models and evaluate their potential benefit in the clinical setting.

---

## [Referee Report · Reviewer #2 (Public Review)]

Summary:

This manuscript by Latini et al describes a methodology to develop Boolean-based predictive logic models that can be applied to uncover altered protein/signalling networks in cancer cells and discover potential new therapeutic targets. As a proof-of-concept, they have implemented their strategy on a hematopoietic cell line engineered to express one of two types of FLT3 internal tandem mutations (FLT3-ITD) found in patients, FLT3-ITD-TKD (which are less sensitive to tyrosine kinase inhibitors/TKIs) and FLT3-ITD-JMD (which are more sensitive to TKIs).

Strengths:

This useful work could potentially represent a step forward towards personalised targeted therapy, by describing a methodology using Boolean-based predictive logic models to uncover altered protein/signalling networks within cancer cells.

Authors have validated their approach by analysing independent, real-world data

Weaknesses:

No weaknesses were observed by this reviewer for the revised version.

---

## [Referee Report · Reviewer #3 (Public Review)]

Summary: The paper "Unveiling the signaling network of FLT3-ITD AML improves drug sensitivity prediction" reports the combination of prior knowledge signaling networks, multiparametric cell-based data on the activation status of 14 crucial proteins emblematic of the cell state downstream of FLT3 obtained under a variety of perturbation conditions and Boolean logic modeling, to gain mechanistic insight into drug resistance in acute myeloid leukemia patients carrying the internal tandem duplication in the FLT3 receptor tyrosine kinase and predict drug combinations that may reverse pharmacoresistant phenotypes. Interestingly, the utility of the approach was validated in vitro and using real-world data.

Strengths:

The model predictions have been validated in vitro and using external data.

This is a complex study, but readability is enhanced by the inclusion of a section that summarizes the study design, plus relevant figures. The availability of data as supplementary material and the availability of code in GitHub are also high points.

Weaknesses:

There are some apparent discrepancies between predicted and observed data that have been seemingly overlooked.

---

## [Author Response · None]

**Author Response**

The following is the authors’ response to the original reviews.

**eLife Assessment**
This useful study could potentially represent a step forward towards personalized medicine by combining cell-based data and a prior-knowledge network to derive Boolean-based predictive logic models to uncover altered protein/signaling networks within cancer cells. However, the level of evidence supporting the conclusions is inadequate, and further validation of the reported approach is required. If properly validated, these findings could be of interest to medical biologists working in the field of cancer and would inform drug development and treatment choices in the field of oncology.

We thank the editor and the reviewer for their constructive comments, which helped us to improve our story. We have now performed new analyses and experiments to further support our proposed approach.

**Public Reviews:**

**Reviewer #1 (Public Review):**
(1) The authors deploy a combination of their own previously developed computational methods and databases (SIGNOR and CellNOptR) to model the FLT3 signaling landscape in AML and identify synergistic drug combinations that may overcome the resistance AML cells harboring ITD mutations in the TKI domain of FLT3 to FLT3 inhibitors. I did not closely evaluate the details of these computational models since they are outside of my area of expertise and have been previously published. The manuscript has significant issues with data interpretation and clarity, as detailed below, which, in my view, call into question the main conclusions of the paper.

The authors train the model by including perturbation data where TKI-resistant and TKIsensitive cells are treated with various inhibitors and the activity (i.e. phosphorylation levels) of the key downstream nodes are evaluated. Specifically, in the Results section (p. 6) they state "TKIs sensitive and resistant cells were subjected to 16 experimental conditions, including TNFa and IGF1 stimulation, the presence or absence of the FLT3 inhibitor, midostaurin, and in combination with six small-molecule inhibitors targeting crucial kinases in our PKN (p38, JNK, PI3K, mTOR, MEK1/2 and GSK3)". I would appreciate more details on which specific inhibitors and concentrations were used for this experiment. More importantly, I was very puzzled by the fact that this training dataset appears to contain, among other conditions, the combination of midostaurin with JNK inhibition, i.e. the very combination of drugs that the authors later present as being predicted by their model to have a synergistic effect. Unless my interpretation of this is incorrect, it appears to be a "self-fulfilling prophecy", i.e. an inappropriate use of the same data in training and verification/test datasets.

We thank the reviewer for this comment. We have now extensively revised the Figure 2B and edited the text to clarify and better describe the experimental conditions of our multiparametric analysis. As the reviewer stated, we have used different combinations of drugs, including midostaurin and JNK inhibitor to generate two cell-specific predictive models recapitulating the main signal transduction events, down-stream FLT3, occurring in resistant (FLT3ITD-TKD) and sensitive (FLT3ITD-JMD) cells. These experiments were performed by treating cells at very early time points to obtain a picture of the signaling response of FLT3-ITD positive cells. Indeed, we have measured the phosphorylation level of signaling proteins, because at these early time points (90 minutes) we do not expect a modulation of downstream crucial phenotypes, including apoptosis or proliferation. To infer perturbations impacting the apoptosis or proliferation phenotypes, we applied a computational two-steps strategy:

(1) We extracted key regulators of ‘apoptosis’ and ‘proliferation’ hallmarks from SIGNOR database.

(2) We applied our recently developed ProxPath algorithm to retrieve significant paths linking nodes of our two optimized models to ‘proliferation’ and ‘apoptosis’ phenotypes.

This allowed us to evaluate in silico the “proliferation” and “apoptosis” rate upon inactivation of each node of the network. With the proposed approach, we identified JNK as a potential drug target to use in combination with FLT3 to restore sensitivity (i.e. in silico inducing apoptosis and reducing proliferation) of FLT3 ITD-TKD cells. We here want to stress once more that although the first piece of information (the effect of JNK and FLT3 inhibition) on sentinel readouts was provided in the training dataset, the second piece of information (the effect on this treatment over the entire model and, as a consequence, on the cellular phenotype) was purely the results of our computational models. As such, we hope that the reviewer will agree that this could not represent a “self-fulfilling prophecy".

That said, we understand that this aspect was not clearly defined in the manuscript. For this reason, we have now (1) extensively revised the Figure 2B; (2) edited the text (pg. 6) to clarify the purpose and the results of our approach; and (3) described in further detail (pg. 16-18) the experimental conditions of our multiparametric analysis.

(2) My most significant criticism is that the proof-of-principle experiment evaluating the combination effects of midostaurin and SP600125 in FLT3-ITD-TKD cell line model does not appear to show any synergism, in my view. The authors' interpretation of the data is that the addition of SP600125 to midostaurin rescues midostaurin resistance and results in increased apoptosis and decreased viability of the midostaurin-resistant cells. Indeed, they write on p.9: "Strikingly, the combined treatment of JNK inhibitor (SP600125) and midostaurin (PKC412) significantly increased the percentage of FLT3ITD-TKD cells in apoptosis (Fig. 4D). Consistently, in these experimental conditions, we observed a significant reduction of proliferating FLT3ITD- TKD cells versus cells treated with midostaurin alone (Fig. 4E)." However, looking at Figs 4D and 4E, it appears that the effects of the midostaurin/SP600125 combination are virtually identical to SP600125 alone, and midostaurin provides no additional benefit. No p-values are provided to compare midostaurin+SP600125 to SP600125 alone but there seems to be no appreciable difference between the two by eye. In addition, the evaluation of synergism (versus additive effects) requires the use of specialized mathematical models (see for example Duarte and Vale, 2022). That said, I do not appreciate even an additive effect of midostaurin combined with SP600125 in the data presented.

We agree with the reviewer that the JNK inhibitor and midostaurin do not have neither a synergic nor additive effect and we have now revised the text accordingly. It is highly discussed in the scientific community whether FLT3ITD-TKD AML cells benefit from midostaurin treatments. In a recently published retroprospective study of K. Dohner et al. (Rücker et al., 2022), the authors investigated the prognostic and predictive impact of FLT3-ITD insertion site (IS) in 452 patients randomized within the RATIFY trial, which evaluated midostaurin additionally to intensive chemotherapy. Their study clearly showed that “Midostaurin exerted a significant benefit only for JMDsole” patients. In agreement with this result, we have demonstrated that midostaurin treatment had no effects on apoptosis of blasts derived from FLT3ITD-TKD patients (Massacci et al., 2023). On the other hand, we and others observed that midostaurin triggers apoptosis in FLT3ITD-TKD cells to a lesser extent as compared to FLT3ITDJMD cells (Arreba-Tutusaus et al., 2016). The data presented here (Fig. 4) and our previously published papers (Massacci et al., 2023; Pugliese et al., 2023) pinpoint that hitting cell cycle regulators (WEE1, CDK7, JNK) induce a significant apoptotic response of TKI resistant FLT3ITD-TKD cells. Prompted by the reviewer comment, we have now revised the text and discussion (pg.9; 14) highlighting the crucial role of JNK in apoptosis induction.

(3) In my view, there are significant issues with clarity and detail throughout the manuscript. For example, additional details and improved clarity are needed, in my view, with respect to the design and readouts of the signaling perturbation experiments (Methods, p. 15 and Fig 2B legend). For example, the Fig 2B legend states: "Schematic representation of the experimental design: FLT3 ITD-JMD and FLT3 ITD-JMD cells were cultured in starvation medium (w/o FBS) overnight and treated with selected kinase inhibitors for 90 minutes and IGF1 and TNFa for 10 minutes. Control cells are starved and treated with PKC412 for 90 minutes, while "untreated" cells are treated with IGF1 100ng/ml and TNFa 10ng/ml with PKC412 for 90 minutes.", which does not make sense to me. The "untreated" cells appear to be treated with more agents than the control cells. The logic behind cytokine stimulation is not adequately explained and it is not entirely clear to me whether the cytokines were used alone or in combination. Fig 2B is quite confusing overall, and it is not clear to me what the horizontal axis (i.e. columns of "experimental conditions", as opposed to "treatments") represents. The Method section states "Key cell signaling players were analyzed through the X-Map Luminex technology: we measured the analytes included in the MILLIPLEX assays" but the identities of the evaluated proteins are not given in the Methods. At the same time, the Results section states "TKIs sensitive and resistant cells were subjected to 16 experimental conditions" but these conditions do not appear to be listed (except in Supplementary data; and Fig 2B lists 9 conditions, not 16). In my subjective view, the manuscript would benefit from a clearer explanation and depiction of the experimental details and inhibitors used in the main text of the paper, as opposed to various Supplemental files/Figures. The lack of clarity on what exactly were the experimental conditions makes the interpretation of Fig 2 very challenging. In the same vein, in the PCA analysis (Fig 2C) there seems to be no reference to the cytokine stimulation status while the authors claim that PC2 stratifies cells according to IGF1 vs TNFalpha. There are numerous other examples of incomplete or confusing legends and descriptions which, in my view, need to be addressed to make the paper more accessible.

We thank the reviewer for his/her comment. We have now extensively revised the text of the manuscript (pg. 6), revised Fig. 2B (now Fig 2C) and methods (pg. 16-18) to improve the clarity of our manuscript, making the take-home messages more accessible. We believe that the revised versions of text and of Figure 2 better explain our strategy and clarify the experimental set up, we added details on the choices of the experimental conditions, and we proposed a better graphic representation of the analysis.

(4) I am not sure that I see significant value in the patient-specific logic models because they are not supported by empirical evidence. Treating primary cells from AML patients with relevant drug combinations would be a feasible and convincing way to validate the computational models and evaluate their potential benefit in the clinical setting.

We thank the reviewer for this comment. We have now performed additional experiments in a small cohort of FLT3-ITD positive patient-derived primary blasts. Specifically, we have treated blasts from 2 FLT3ITD-TKD patients and 3 FLT3ITD-JMD+TKD patients with PKC412 (100nM) 24h and/or 10μM SP600125 (JNK inhibitor). After 24h of treatment we have measured the apoptotic rate. As shown below and in the new Fig. 4F (see pg.10, main text), midostaurin triggers higher levels of apoptosis in FLT3ITD-JMD+TKD blasts as compared to FLT3ITD-TKD blasts. Importantly, treatment with the JNK inhibitor SP600125 alone triggers apoptosis in FLT3ITD-TKD blasts, validating the crucial role of JNK in FLT3ITD-TKD cell survival and TKI resistance. The combined treatment of midostaurin and SP600125 increases the percentage of apoptotic cells as compared to midostaurin treatment alone but to a lesser extent than single agent treatment. This result is in agreement with the current debate in the scientific community on the actual beneficial effect of midostaurin treatment in FLT3ITD-TKD AML patients.

**Author response image 1. sa4fig1:** Primary samples from AML patients with the FLT3ITD-TKD mutation (n=2, yellow bars) or the FLT3ITD-JMD/TKD mutation (n=3, blue bars) were exposed to Midostaurin (100nM, PKC412), and JNK inhibitor (10µM, SP600125) for 48 hours, or combinations thereof. The specific cell death of gated AML blasts was calculated to account for treatment-unrelated spontaneous cell death. The bars on the graph represent the mean values with standard errors.

**Reviewer #2 (Public Review):**
Summary:This manuscript by Latini et al describes a methodology to develop Boolean-based predictive logic models that can be applied to uncover altered protein/signalling networks in cancer cells and discover potential new therapeutic targets. As a proof-of-concept, they have implemented their strategy on a hematopoietic cell line engineered to express one of two types of FLT3 internal tandem mutations (FLT3-ITD) found in patients, FLT3-ITD-TKD (which are less sensitive to tyrosine kinase inhibitors/TKIs) and FLT3-ITD-JMD (which are more sensitive to TKIs).Strengths:This useful work could potentially represent a step forward towards personalised targeted therapy, by describing a methodology using Boolean-based predictive logic models to uncover altered protein/signalling networks within cancer cells. However, the weaknesses highlighted below severely limit the extent of any conclusions that can be drawn from the results.Weaknesses:While the highly theoretical approach proposed by the authors is interesting, the potential relevance of their overall conclusions is severely undermined by a lack of validation of their predicted results in real-world data. Their predictive logic models are built upon a set of poorlyexplained initial conditions, drawn from data generated in vitro from an engineered cell line, and no attempt was made to validate the predictions in independent settings. This is compounded by a lack of sufficient experimental detail or clear explanations at different steps. These concerns considerably temper one's enthusiasm about the conclusions that could be drawn from the manuscript.

We thank the reviewer for the thorough review and kind comments about our manuscript. We hope the changes and new data we provide further strengthen it in his or her eyes.

Some specific concerns include:(1) It remains unclear how robust the logic models are, or conversely, how affected they might be by specific initial conditions or priors that are chosen. The authors fail to explain the rationale underlying their input conditions at various points. For example: - at the start of the manuscript, they assert that they begin with a pre-PKN that contains "76 nodes and 193 edges", though this is then ostensibly refined with additional new edges (as outlined in Fig 2A). However, why these edges were added, nor model performance comparisons against the basal model are presented, precluding an evaluation of whether this model is better.

We understand the reviewer’s concern. We have now complemented the manuscript with an extended version of the proposed modelling strategy offering a detailed description of the pipeline and the rationale behind each choice (Supplementary material, pg.14-19). Furthermore, we also referenced the manuscript to a GitHub repository where users can follow and reproduce each step of the pipeline (https://github.com/SaccoPerfettoLab/FLT3ITD_driven_AML_Boolean_models).

At a later step (relevant to Fig S4 and Fig 3), they develop separate PKNs, for each of the mutation models, that contain "206 [or] 208 nodes" and "756 [or] 782 edges", without explaining how these seemingly arbitrary initial conditions were arrived at. Their relation to the original parameters in the previous model is also not investigated, raising concerns about model over-fitting and calling into question the general applicability of their proposed approach. The authors need to provide a clearer explanation of the logic underlying some of these initial parameter selections, and also investigate the biological/functional overlap between these sets of genes (nodes).

We thank the reviewer for raising this question. Very briefly, the proposed optimization strategy falls in a branch of the modelling, where the predictive model is, indeed, driven by the data (Blinov and Moraru, 2012). From a certain point of view, the scope of optimization is the one of fitting the experimental data in the best way possible. To achieve this, we followed standard practices (Dorier et al., 2016; Traynard et al., 2017). To address the issue of “calling into question the general applicability of their proposed approach”, we have compared the activity status of nodes in the models with ‘real data’ extracted from cell lines and patients’ samples to reassure about the robustness and scalability of the strategy (please see below, response to point 3 pg. 9).

Finally, as mentioned in the previous point, we have now provided a detailed supplementary material, where we have described all the aspects mentioned by the reviewer: step-by-step changes in the PKN, the choice of the parameters and other details can be traced over the novel text and are also available in the GitHub repository (https://github.com/SaccoPerfettoLab/FLT3-ITD_driven_AML_Boolean_models).

(2) There is concern about the underlying experimental data underpinning the models that were generated, further compounded by the lack of a clear explanation of the logic. For example, data concerning the status of signalling changes as a result of perturbation appears to be generated from multiplex LUMINEX assays using phosphorylation-specific antibodies against just 14 "sentinel" proteins. However, very little detail is provided about the rationale underlying how these 14 were chosen to be "sentinels" (and why not just 13, or 15, or any other number, for that effect?). How reliable are the antibodies used to query the phosphorylation status? What are the signal thresholds and linear ranges for these assays, and how would these impact the performance/reliability of the logic models that are generated from them?

We thank the reviewer for this comment as it gives us the opportunity to clarify and better explain the criteria behind the experimental data generation.

Overall, we revised the main text at page 6 and the Figure 2B to improve the clarity of our experimental design. Specifically, the sentinels were chosen because they were considered indirect or direct downstream effectors of the perturbations and were conceived to serve as both a benchmarking system of the study and a readout of the global perturbation of the system. To clarify this aspect, we have added a small network (compressed PKN) in Figure 2B to show that the proteins (green nodes) we chose to measure in the LUMINEX multiplex assay are “sentinels” of the activity of almost all the pathways included in the Prior knowledge network. Moreover, we implemented the methods section “Multiparametric experiment of signaling perturbation” (pg. 16-18), where we added details about the antibodies used in the assay paired with the target phosphosites and their functional role (Table 3). We also better specified the filtering process based on the number of beads detected per each antibody used (pg. 18). About the reliability of the measurements, we can say that the quality of the perturbation data impacts greatly on the logic models’ performance. xMAP technology been already used by the scientific community to generate highly reproducible and reliable multiparametric dataset for model training (Terfve et al., 2012). Additionally, we checked that for each sentinel we could measure a fully active state, a fully inactive state and intermediate states. Modulation of individual analytes are displayed in Figure S3.

**Author response image 2. sa4fig2:** Partial Figure of normalization of analytes activity through Hill curves. Experimental data were normalized and scaled from 0 to 1 using analyte-specific Hill functions. Raw data are reported as triangles, normalized data and squares. Partial Figure representing three plots of the FLT3 ITD-JMD data (Complete Figure in Supplementary material Fig S3).

(3) In addition, there are publicly available quantitative proteomics datasets from FLT3-mutant cell lines and primary samples treated with TKIs. At the very least, these should have been used by the authors to independently validate their models, selection of initial parameters, and signal performance of their antibody-based assays, to name a few unvalidated, yet critical, parameters. There is an overwhelming reliance on theoretical predictions without taking advantage of real-world validation of their findings. For example, the authors identified a set of primary AML samples with relevant mutations (Fig 5) that could potentially have provided a valuable experimental validation platform for their predictions of effective drug combination. Yet, they have performed Boolean simulations of the predicted effects, a perplexing instance of adding theoretical predictions on top of a theoretical prediction!

Additionally, there are datasets of drug sensitivity on primary AML samples where mutational data is also known (for example, from the BEAT-AML consortia), that could be queried for independent validation of the authors' models.

We thank the reviewer for this comment that helped us to significantly strengthen our story. Prompted by his/her comment, we have now queried three different datasets for independent validation of our logic models. Specifically, we have taken advantage of quantitative phosphoproteomics datasets of FLT3-ITD cell lines treated with TKIs (Massacci et al., 2023), phosphoproteomic data of FLT3-ITD positive patients-derived primary blast (Kramer et al., 2022) and of drug sensitivity data on primary FLT3-ITD positive AML samples (BEAT-AML consortia)

Comparison with phosphoproteomic data of FLT3-ITD cell lines treated with TKIs (Massacci et al., 2023)

Here, we compared the steady state of our model upon FLT3 inhibition with the phosphoproteomic data describing the modulation of 16,319 phosphosites in FLT3-ITD BaF3 cells (FLT3ITD-TKD and FLT3ITD-JMD) upon TKI treatment (i.e. quizartinib, a highly selective FLT3 inhibitor). As shown in the table below and new Figure S5A, the activation status of the nodes in the two generated models is highly comparable with the level of regulatory phosphorylations reported in the reference dataset. Briefly, to determine the agreement between each model and the independent dataset, we focused on the phosphorylation level of specific residues that (i) regulate the functional activity of sentinel proteins (denoted in the ‘Mode of regulation’ column) and (ii) that were measured in this work to train the model. So, we cross-referenced the sentinel protein status in FLT3 inhibition simulation (as denoted in the 'Model simulation of FLT3 inhibition' column) with the functional impact of phosphorylation measured in Massacci et. al dataset (as denoted in the 'Functional impact in quizartinib dataset' column). Points of congruence were summarized in the 'Consensus' column. As an example, if the phosphorylation level of an activating residue decreases (e.g., Y185 of Mapk1), we can conclude that the protein is inhibited (‘Down-reg’) and this is coherent with model simulation in which Mapk1 is ‘Inactive’.

**Author response image 3. sa4fig3:** 

Comparison with phosphoproteomic data of FLT3-ITD patient-derived primary blasts (Kramer et al., 2022)

Using the same criteria, we extended our validation efforts by comparing the activity status of the proteins in the “untreated” simulation (i.e. reproducing the tumorigenic state where FLT3, IGF1R and TNFR are set to be active) with their phosphorylation levels in the dataset by Kramer et al. (Kramer et al., 2022). Briefly, this dataset gathers phosphoproteomic data from a cohort of 44 AML patients and we restricted the analysis to 11 FLT3-ITD-positive patients. Importantly, all patients carry the ITD mutation in the juxta membrane domain (JMD), thus allowing for the comparison with FLT3 ITD-JMD specific Boolean model, exclusively.

The results are shown in the heatmap below. Each cell in the heatmap reports the phosphorylation level of sentinel proteins’ residues in the indicated patient (red and blue indicate up- or- down-regulated phosphoresidues, respectively). Patients were clustered according to Pearson correlation. We observed a good level of agreement between the patients’ phosphoproteomics data and our model (reported in the column “Tumor simulation steady state”) for a subset of patients highlighted within the black rectangle. However, for the remaining patients, the level of agreement is poor. The main reason is that our work focuses on FLT3-ITD signaling and a systematic translation of the Boolean modeling approach to the entire cohort of AML patients would require the inclusion of the impact of other driver mutations in the network. This is actually a current and a future line of investigation of our group. We have revised the discussion, taking this result into consideration.

**Author response image 4. sa4fig4:** 

Comparison with drug sensitivity data on primary FLT3-ITD positive AML samples (BEAT-AML consortia)

Here we took advantage of the Beat AML programme on a cohort of 672 tumour specimens collected from 562 patients. The BEAT AML consortium provides whole-exome sequencing, RNA sequencing and analyses of ex vivo drug sensitivity of this large cohort of patient-derived primary blasts. We focused on drug sensitivity screening on 134 patients carrying the typical FLT3-ITD mutation in the JMD region. Unfortunately, the ITD insertion in the TKD region is less characterized and additional in-depth sequencing studies are required to identify in this cohort FLT3ITD-TKD positive blasts. Next, we focused on those compounds hitting nodes present in the FLT3ITD-JMD Boolean model. Specifically, we selected drugs inhibiting FLT3, PI3K, mTOR, JNK and p38 and we calculated the average IC50 of FLT3ITD-JMD patient-derived primary blasts for each drug. These results are reported as a bar graph in the new Fig. S5B and below (upper panel) and were compared with the apoptotic and proliferation rate measured in silico simulation of the FLT3ITD-JMD Boolean model. Drug sensitivity screening on primary FLT3ITD-JMD blasts revealed that inhibition of FLT3, PI3K and mTOR induces cell death at low drug concentrations in contrast with JNK and p38 inhibitors showing higher IC50 values. These observations are consistent with our simulation results of the FLT3ITD-JMD model. As expected, in silico inhibition of FLT3 greatly impacts apoptosis and proliferation. Additionally, in silico suppression of mTOR and to a lesser extent PI3K and p38 affect apoptosis and proliferation. Of note, JNK inhibition neither in silico nor in vitro seems to affect viability of FLT3ITD-JMD cells.

**Author response image 5. sa4fig5:** 

Altogether these publicly available datasets independently validate our models, strengthening the reliability and robustness of our approach.

We have now revised the main text (pg. 8; 9) and added a new Figure (Fig. S5) in the supplementary material; we collected the results of the analysis in TableS6.

(4) There are additional examples of insufficient experimental detail that preclude a fuller appreciation of the relevance of the work. For example, it is alluded that RNA-sequencing was performed on a subset of patients, but the entire methodological section detailing the RNA-seq amounts to just 3 lines! It is unclear which samples were selected for sequencing nor where the data has been deposited (or might be available for the community - there are resources for restricted/controlled access to deidentified genomics/transcriptomics data).

We apologize for the lack of description regarding the RNA sequencing of patient samples. We have now added details of this approach in the method section (pg. 24), clearly explained in text how we selected the patients for the analysis. Additionally, data has now been deposited in the GEO database (accession number: GSE247483).

The sentences we have rephrased are below:

“We analyzed the mutational and expression profiles of 262 genes (Table S7), relevant to hematological malignancies in a cohort of 14 FLT3-ITD positive de novo AML patients (Fig. 5A, panel a). Since, follow-up clinical data were available for 10 out of 14 patients (Fig. 5B, Table S9), we focused on this subset of patients. Briefly, the classification of these 10 patients according to their ITD localization (see Methods) was as follows: 8 patients with FLT3ITD-JMD, 4 with FLT3ITD-JMD+TKD, and 2 with FLT3ITD-TKD (Fig. 5A, panel b). The specific insertion sites of the ITD in the patient cohort are shown in Table S8.

Similarly, in the "combinatory treatment inference" methods, it states "...we computed the steady state of each cell line best model....." and "Then we inferred the activity of "apoptosis" and "proliferation" phenotypes", without explaining the details of how these were done. The outcomes of these methods are directly relevant to Fig 4, but with such sparse methodological detail, it is difficult to independently assess the validity of the presented data.Overall, the theoretical nature of the work is hampered by real-world validation, and insufficient methodological details limit a fuller appreciation of the overall relevance of this work.

We thank the reviewer for the insightful feedback regarding the methodology in our paper.

About ‘real-world validation’ we have extensively replied to this issue in point 3 (pg. 9-14 of this document). For what concerns the ‘insufficient methodological details’, we have made substantial improvements to enhance clarity and reproducibility, that encompass: (i) revisions in the main text and in the Materials and Methods section; (ii) detailed explanation of each step and decisions taken that can be accessed either as an extended Materials and Methods section (Supplementary material, pg. 14-19) and through our GitHub repository (https://github.com/SaccoPerfettoLab/FLT3-ITD_driven_AML_Boolean_models). We sincerely hope this addition addresses concerns and facilitates a more thorough and independent assessment of our work.

**Reviewer #3 (Public Review):**
Summary:The paper "Unveiling the signaling network of FLT3-ITD AML improves drug sensitivity prediction" reports the combination of prior knowledge signaling networks, multiparametric cell-based data on the activation status of 14 crucial proteins emblematic of the cell state downstream of FLT3 obtained under a variety of perturbation conditions and Boolean logic modeling, to gain mechanistic insight into drug resistance in acute myeloid leukemia patients carrying the internal tandem duplication in the FLT3 receptor tyrosine kinase and predict drug combinations that may reverse pharmacoresistant phenotypes. Interestingly, the utility of the approach was validated in vitro, and also using mutational and expression data from 14 patients with FLT3-ITD positive acute myeloid leukemia to generate patient-specific Boolean models.Strengths:The model predictions were positively validated in vitro: it was predicted that the combined inhibition of JNK and FLT3, may reverse resistance to tyrosine kinase inhibitors, which was confirmed in an appropriate FLT3 cell model by comparing the effects on apoptosis and proliferation of a JNK inhibitor and midostaurin vs. midostaurin alone.Whereas the study does have some complexity, readability is enhanced by the inclusion of a section that summarizes the study design, plus a summary Figure. Availability of data as supplementary material is also a high point.

We thank the reviewer for his/her constructive comments about our manuscript. We believe that our story has been significantly strengthened by the changes and new data we provided.

Weaknesses:(1) Some aspects of the methodology are not properly described (for instance, no methodological description has been provided regarding the clustering procedure that led to Figs. 2C and 2D).

We apologize for the lack of proper description of the methodology. We have extensively revised the methods section and worked to improve the clarity. We have now added a description of the clustering procedures in the methods section (pg. 19) of new Fig. S2D., Fig.S2E.

It is not clear in the manuscript whether the patients gave their consent to the use of their data in this study, or the approval from an ethical committee. These are very important points that should be made explicit in the main text of the paper.

We thank the reviewer for this comment. We have now added the following sentence (pg. 24):“Peripheral blood (PB) samples from 14 AML patients were obtained upon patient’s informed consent.”

The authors claim that some of the predictions of their models were later confirmed in the follow-up of some of the 14 patients, but it is not crystal clear whether the models helped the physicians to make any decisions on tailored therapeutic interventions, or if this has been just a retrospective exercise and the predictions of the models coincide with (some of) the clinical observations in a rather limited group of patients. Since the paper presents this as additional validation of the models' ability to guide personalized treatment decisions, it would be very important to clarify this point and expand the presentation of the results (comparison of observations vs. model predictions).

As described in the introduction section, this study was inspired by an urgent clinical problem in AML research: patients carrying the ITD in the TKD domain of the FLT3 receptor display poor prognosis and do not respond to current therapy: Midostaurin (which on the other hand is effective in patients with the ITD in the JMD domain).

To fill this gap, we gathered a team of 18 participants, of which 7 have a clinical background and have expertise in the diagnosis, treatment and management of AML patients and 5 are experts in Boolean modeling. The scope of the project is the development of a computational approach to identify possible alternative solutions for FLT3ITD-TKD AML patients, generating future lines of investigations. Drug combinations are currently under investigation as a potential means of avoiding drug resistance and achieving more effective and durable treatment responses. However, it is impractical to test for potential synergistic properties among all available drugs using empirical experiments alone. With our approach, we developed models that recreated in silico the main differences in the signaling of sensitive and resistant cells to support the prioritization of novel therapies. Prompted by the reviewer suggestions, we have now extended the validation of our models, through the comparison with publicly available cell lines and patient-derived dataset. We have also confirmed our results by performing in vitro experiments in patient-derived primary blasts treated with midostaurin and/or JNK inhibitor. Importantly, we have already demonstrated that hitting cell cycle regulators in FLT3ITD-TKD cells can be an effective approach to kill resistant leukemia cells (Massacci et al., 2023; Pugliese et al., 2023). We are aware that changing the clinical practice and the therapies for patients require a proper clinical study which goes far beyond the scope of this manuscript.

However, we hope that our results can be translated soon from “bench-to-bed”. Importantly, we believe that our study can open lines of investigations aimed at the application of our approach to identify promising therapeutic strategies in other clinical settings.

**Recommendations for the authors**
The reviewers have highlighted significant issues regarding the inadequate level of evidence to support some of the conclusions, plus lack of an exhaustive methodological description that may jeopardize reproducibility.

We hope that the editor and the reviewers will appreciate the extensive revision we made and new data and analysis we provided to strengthen our story.

**Reviewer #1 (Recommendations For The Authors):**
(1) In Fig 2D the hierarchical tree is off-set in relation to the treatment symbols and names in the middle of the Figure. In addition, I do not see FLT3i combination with JNKi in the JMD cells (perhaps, a coloring error?).

We thank the reviewer for this observation. We have now revised the hierarchical tree, which is now in Figure S2D, we have aligned the tree with the symbols and names and corrected the colouring error for the sample FLT3i+JNKi in JMD cells.

(2) Midostaurin and PKC412 refer to the same drug and are used interchangeably in the manuscript. Using one name consistently would improve readability.

We have now improved the readability of the text and the Figures by choosing “Midostaurin” when we refer to the FLT3 inhibitor.

(3) It is not clear to me why the FLT3-ITD-JMD cells are not presented in Fig. 4B. Perhaps their values are 0? In that case, the readability would be improved by including a thin blue line representing zero values. Additionally, on p.8 the authors state "Interestingly, in the FLT3ITDTKD model, the combined inhibition of JNK and FLT3, exclusively, in silico restores the TKI sensitivity, as revealed by the evaluation of the apoptosis and proliferation levels (Fig. 4B-C)." but Fig. 4C shows no differential effects of JNK inhibition in sensitive versus resistant cells.

To address the reviewer's point, we’ve added a thin blue line representing the zero values of the FLT3ITD-JMD in the results of the simulations in Figure 4B. Regarding the Figure 4C, the reviewer is right in saying that there is no difference in terms of proliferation between sensitive and resistant cells upon JNKi and FLT3i co-inhibition. However, we can see lower proliferation levels in both cell lines as compared to the “untreated” condition. Indeed, the simulation suggests that by combining JNK and FLT3 inhibition we restore the resistant phenotype lowering the proliferation rate of the resistant cells to the TKI-sensitive levels.

**Reviewer #2 (Recommendations For The Authors):**
I have addressed a number of concerns in the public review. Much better effort needs to be made to provide sufficient methodological detail (to permit independent validation by a sufficiently capable and motivated party) and explain the rationale of important parameter selections. Furthermore, I urge the authors to take advantage of the plethora of publicly available real-world data to validate their predicted outcomes.

We are grateful to the reviewer for the careful revisions. All the aspects raised have been discussed in the specific sections of the public review. In summary, we have provided more methodological details, by revising the text, the methods session, by adding a new step-by-step description of the modelling strategy, the parameters and the criteria adopted in each phase (supplementary methods) and by referring to the entire code developed. Prompted by the reviewer suggestions, we have performed a novel and extensive comparison of our model with three different publicly available datasets. This analysis significantly strengthens our story, and a new supplementary Figure (Fig. S5) summarizes our findings (pg. 9-14 of this document).

**Reviewer #3 (Recommendations For The Authors):**
(1) At first sight, the distribution of the data points in the PCA space does not really seem to speak of nice clustering. Have the authors computed any clustering validation metric to assess if their clustering strategy is adequate and how informative the results are? Further analysis of this point of the article is precluded by the absence of a clear methodological description.

Here we have used the PCA analysis to obtain a global view of our complex multiparametric data. We have now worked on the PCA to improve its readability. As shown in the new Figure 2D, PCA analysis showed that the activity level of sentinel proteins stratifies cells according to FLT3 activation status (component 1: presence vs absence of FLT3i) and cytokine stimulation (component 2: IGF1 vs TNF⍺). We have now added new experimental details on this part in the methods section (pg. 19) and we deposited the code used for the clustering strategy on the GitHub repository (https://github.com/SaccoPerfettoLab/FLT3ITD_driven_AML_Boolean_models).

(2) Whereas scientists and medical professionals who work in the field of oncology may be familiar with some of the abbreviations used here, it would be good for improved readability by a more general audience to make sure that all the abbreviations (e.g., TKI) are properly defined the first time that they appear in the text.

We thank the reviewer for this observation. To improve the readability of the text, we properly defined all the abbreviations in their first appearance, and we added the “Abbreviation” paragraph at page 15 of the manuscript to summarize them all.

(3) How were the concentrations of the combined treatments chosen in the cell assays used as validation?

We thank the reviewer for giving us the chance to clarify this point. We implemented the Methods with additional information about the treatments used in the validations. We detailed the SP600125 IC50 evaluation and usage in our cell lines (pg.22): IC50 values are approximately 1.5 µM in FLT3-ITD mutant cell lines; the SP600125 treatment affects cell viability, reaching a plateau phase of cell death and at about 2 µM. I used the minimal dose of SP600125 (10µM) to properly inhibit JNK. (Kim et al., 2010; Moon et al., 2009).

We also specified (pg.22) that the concentration of Midostaurin was chosen based on the previously published work (Massacci et al., 2022): FLT3 ITD-TKD cells treated with Midostaurin 100nM show lower apoptotic rate and higher cell viability compared to FLT3 ITD-JMD cells.

The concentration of SB203580 and UO126 was chosen based on previous data available in the lab and set up experiments (pg.22).

(4) The authors say that "we were able to derive patient-specific signaling features and enable the identification of potential tailored treatments restoring TKI resistance" and that "our predictions were confirmed by follow-up clinical data for some patients". However, the results section on this part of the manuscript is rather scarce (the main text should be much more descriptive about the results summarized in Fig. 5, which are not self-explanatory).

We thank the reviewer for this observation. We have now expanded the text to provide a more comprehensive description of the results about personalized Boolean model generation and usage and the content presented in Fig. 5 (pg.10-12).

(5) I do not really agree with the final conclusion about this paper being "the proof of concept that our personalized informatics approach described here is clinically valid and will enable us to propose novel patient-centered targeted drug solutions". First, the clinical data used here belongs to a rather low number of patients. Second, as mentioned before, it is not clear if the models have been used to make any prospective decision or if this conclusion is drawn from an in vitro assay plus a retrospective analysis on a limited number of patients. Moreover, a description of the results and the discussion of the part of the manuscript dealing with patientspecific models is rather scarce, and it is difficult to see how the authors support their conclusions. Also, the statement " In principle, the generalization of our strategy will enable to obtain a systemic perspective of signaling rewiring in different cancer types, driving novel personalized approaches" may be a bit overoptimistic if one considers that so far, the approach has only been applied to a single type of drug-resistant cancer.

We thank the reviewer for this comment. We agree with the referees that the clinical data we used belongs to a rather low number of patients. However, during the revision we have extensively worked to support the clinical relevance of our models and our discoveries. Specifically, we have compared our Boolean logic models with two different publicly available datasets on phosphoproteomics and drug sensitivity of FLT3ITD-JMD and FLT3ITD-TKD cell lines and blasts (FigS5 and answer to reviewer 2, point 3). Importantly, these datasets independently validated our models, highlighting that our approach has a translational value. Additionally, we have performed novel experiments by measuring the apoptotic rate of patient-derived primary blasts upon pharmacological suppression of JNK (Fig. 4H, pg. 10 of main text). Our data highlights that our approach has the potential to suggest novel effective treatments.

That said, we have now revised the discussion to avoid overstatements.

References

Arreba-Tutusaus, P., Mack, T.S., Bullinger, L., Schnöder, T.M., Polanetzki, A., Weinert, S., Ballaschk, A., Wang, Z., Deshpande, A.J., Armstrong, S.A., Döhner, K., Fischer, T., Heidel, F.H., 2016. Impact of FLT3-ITD location on sensitivity to TKI-therapy in vitro and in vivo. Leukemia 30, 1220–1225. https://doi.org/10.1038/leu.2015.292

Blinov, M.L., Moraru, I.I., 2012. Logic modeling and the ridiculome under the rug. BMC Biol 10, 92. https://doi.org/10.1186/1741-7007-10-92

Dorier, J., Crespo, I., Niknejad, A., Liechti, R., Ebeling, M., Xenarios, I., 2016. Boolean regulatory network reconstruction using literature based knowledge with a genetic algorithm optimization method. BMC Bioinformatics 17, 410. https://doi.org/10.1186/s12859-016-1287-z

Kramer, M.H., Zhang, Q., Sprung, R., Day, R.B., Erdmann-Gilmore, P., Li, Y., Xu, Z., Helton, N.M., George, D.R., Mi, Y., Westervelt, P., Payton, J.E., Ramakrishnan, S.M., Miller, C.A., Link, D.C., DiPersio, J.F., Walter, M.J., Townsend, R.R., Ley, T.J., 2022. Proteomic and phosphoproteomic landscapes of acute myeloid leukemia. Blood 140, 1533–1548. https://doi.org/10.1182/blood.2022016033

Massacci, G., Venafra, V., Latini, S., Bica, V., Pugliese, G.M., Graziosi, S., Klingelhuber, F., Krahmer, N., Fischer, T., Mougiakakos, D., Boettcher, M., Perfetto, L., Sacco, F., 2023. A key role of the WEE1-CDK1 axis in mediating TKI-therapy resistance in FLT3-ITD positive acute myeloid leukemia patients. Leukemia 37, 288–297. https://doi.org/10.1038/s41375-022-01785-w

Pugliese, G.M., Venafra, V., Bica, V., Massacci, G., Latini, S., Graziosi, S., Fischer, T., Mougiakakos, D., Boettcher, M., Perfetto, L., Sacco, F., 2023. Impact of FLT3-ITD location on cytarabine sensitivity in AML: a network-based approach. Leukemia 37, 1151–1155. https://doi.org/10.1038/s41375-023-01881-5

Rücker, F.G., Du, L., Luck, T.J., Benner, A., Krzykalla, J., Gathmann, I., Voso, M.T., Amadori, S., Prior, T.W., Brandwein, J.M., Appelbaum, F.R., Medeiros, B.C., Tallman, M.S., Savoie, L., Sierra, J., Pallaud, C., Sanz, M.A., Jansen, J.H., Niederwieser, D., Fischer, T., Ehninger, G., Heuser, M., Ganser, A., Bullinger, L., Larson, R.A., Bloomfield, C.D., Stone, R.M., Döhner, H., Thiede, C., Döhner, K., 2022. Molecular landscape and prognostic impact of FLT3-ITD insertion site in acute myeloid leukemia: RATIFY study results. Leukemia 36, 90–99. https://doi.org/10.1038/s41375-021-01323-0

Terfve, C., Cokelaer, T., Henriques, D., MacNamara, A., Goncalves, E., Morris, M.K., van Iersel, M., Lauffenburger, D.A., Saez-Rodriguez, J., 2012. CellNOptR: a flexible toolkit to train protein signaling networks to data using multiple logic formalisms. BMC Syst Biol 6, 133. https://doi.org/10.1186/1752-0509-6-133

Traynard, P., Tobalina, L., Eduati, F., Calzone, L., Saez-Rodriguez, J., 2017. Logic Modeling in Quantitative Systems Pharmacology: Logic Modeling in Quantitative Systems Pharmacology. CPT Pharmacometrics Syst. Pharmacol. 6, 499–511. https://doi.org/10.1002/psp4.12225